

# Surrogate-Based Design Optimization of Floating Wind Turbines in Time Domain

Büsra Yildirim[1], Nikolay Dimitrov[1], Athanasios Kolios[1], and Asger Bech Abrahamsen[1]

[1]DTU Wind and Energy Systems, Frederiksborgvej 399, DK-4000, Roskilde, Denmark

**Correspondence:** Büsra Yildirim (bysyi@dtu.dk)

**Abstract.**

Floating wind turbine (FWT) design involves higher costs and greater uncertainty than onshore or fixed-bottom offshore turbines due to low technology maturity, limited operational experience, and harsh marine environments; these factors have led to conservative design practices. To address these challenges, we introduce a novel two-step deterministic surrogate-based optimization framework that enables efficient time-domain design optimization for FWTs. In the first step, analytical design constraints are applied to refine the design space and establish a feasible region. In the second step, a surrogate model is trained on high-fidelity aero-hydro-elastic simulations, covering the reduced design space defined from step 1. During an optimization run, the surrogate model replaces computationally expensive direct time-domain analyses, capturing the dynamic response of the system with significantly reduced computational effort. This approach effectively balances model fidelity and computational cost, bridging the gap between conceptual and detailed design phases for floating wind structures. We demonstrate the framework on a semisubmersible platform (UMaine VolturnUS) coupled with the IEA 15 MW reference wind turbine, a representative large-scale FWT. Two primary design variables – the buoyancy column diameter and the overall floater radius – are optimized to minimize the levelized cost of energy (LCOE) of the system. The optimization incorporates global structural limit state constraints covering ultimate (ULS), fatigue (FLS), and serviceability (SLS) requirements to ensure the design's structural feasibility. The surrogate-assisted optimization yields a design that achieves a LCOE of 176.9 /MWh, which is a 3.7 % reduction in LCOE relative to the baseline, with feasibility validated against all ULS, FLS, and SLS criteria. These results highlight the framework's potential to reduce FWT costs and improve design reliability by enabling time-domain optimization without excessive computational expense.

## 1 Introduction

Offshore wind energy has emerged as a key element in the global shift towards sustainable energy production, offering significant advantages in wind resource quality and availability compared to traditional onshore installations (Esteban et al., 2011). However, as development expands into deeper waters exceeding 60 meters, floating wind turbine (FWT) systems become indispensable, introducing unique design complexities (Europe, 2017). These floating structures must reliably withstand dynamic interactions from combined aerodynamic, hydrodynamic, structural, and control forces, presenting substantial engineering challenges. Despite their promising potential, current FWT designs face significant barriers, elevated costs, increased



uncertainty, and limited operational experience, largely stemming from the nature of floating offshore technology. Therefore, addressing these challenges through advanced optimization strategies becomes crucial (Backwell et al., 2024).

Design optimization of the engineering system has multiple steps, including preliminary design, conceptual design, and detailed design. Depending on the design stage, the problem definition of the optimization process and the tools used differ in terms of different modeling fidelities. In terms of floating offshore wind turbine systems, different model fidelities are used for each design process. Preliminary/conceptual design optimization procedures usually require simpler models. Such simpler models will not necessarily capture all relevant phenomena and will not be capable of evaluating all relevant limit states. Moving towards more detailed design considering all standard limit states will require more complex models, and the computational time requires attention. A key point of attention regarding modelling fidelity is the trade-off between the frequency domain and time domain models. The dynamics of FWTs require capturing the complex interaction of the system and the stochastic environment. Nonlinear time domain analyses are required to capture these complex interactions in the FWT dynamics, including aerodynamic, hydrodynamic, control, and structural dynamics (Hegseth et al., 2020). Simplified models can be implemented to explore the design space or conduct the conceptual design phase to obtain optimum solutions. To solve this problem, different approaches are implemented in the literature for different design phases and modelling approaches for FWTs are presented in Table 1.

Frequency-domain models require less computational time compared to time-domain models. Frequency-domain approaches are therefore widely preferred in literature for building simplified FWT models and solving design optimization problems in the conceptual design stage. In Karimi et al. (2017), design optimization in the frequency domain is carried out to explore a wide range of platform designs with three stability classes on a linearized 5 MW turbine, whilst a multi-objective genetic algorithm optimization problem is built considering tower top acceleration and the cost of the platform as design performance criteria. Hegseth et al. (2021) investigated the effect of environmental conditions and inspection strategies on long-term fatigue reliability and design optimization using a frequency domain model. From this work, it is concluded that environmental model uncertainties play a significant role in fatigue damage, particularly when uncertainties are included, resulting in a nearly constant damage reduction of approximately two-thirds along the tower and upper part of the platform. A gradient-based design optimization of FWTs is implemented in the frequency domain for a single objective design problem for fast evaluation of the conceptual design stage (Dou et al., 2020).

The next stage of the design optimization requires a detailed design procedure, including time domain simulations or higher fidelity tools. Time domain simulations can provide a more accurate estimation of the loads, but also have higher computational costs. Therefore, it is not directly possible to conduct time domain design optimization without simplifications or reduction in the load scenarios. Leimeister developed an automated design optimization framework (Leimeister et al., 2020a) for global limit states using Modelica and Dymola for an OC3 spar buoy (Leimeister et al., 2020b), resulting in almost a 24% reduction in structural mass of the spar compared to the reference OC3 Spar buoy design considered. In addition to the deterministic approaches, there are a few examples of reliability-based design optimization (RBDO) studies for FWTs in the literature. RBDO methods incorporate uncertainty information into the design process, ensuring simultaneously a safe, robust, and cost-efficient design. The first application of RBDO on FWTs for floater optimization combines a quadratic response surface approach and

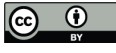

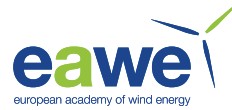 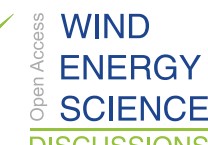

**Table 1.** Modeling approach comparison for FWTs.

| Modeling Approach | Advantages | Disadvantages |
|---|---|---|
| **Frequency-Domain Models** | – Computationally efficient (fast calculations) (Borg and Collu, 2015)<br>– Suitable for preliminary and conceptual design phases<br>– Enables rapid exploration of large design spaces<br>– Facilitates multi-objective optimization with reduced computational demand | – Assumes linear system behavior, limiting accuracy (Borg and Collu, 2015; Journée and Massie, 2001)<br>– Underpredicts dynamic and nonlinear structural responses<br>– Unable to reliably capture transient (Borg and Collu, 2015) and nonlinear interactions (e.g., memory effects) (Journée and Massie, 2001)<br>– Necessitates application of safety factors due to inherent uncertainties (Pillai et al., 2018) |
| **Time-Domain Models** | – High accuracy in capturing nonlinear and transient structural responses<br>– Comprehensive representation of aerodynamic, hydrodynamic, structural, and control interactions (GL, 2018)<br>– Reliable fatigue (GL, 2018) and ultimate load estimations<br>– Appropriate for detailed design stages and verification purposes | – Significant computational expense (Borg and Collu, 2015)<br>– Require input data from frequency domain simulations (Borg and Collu, 2015)<br>– Not directly practical for iterative design optimization processes (Pegalajar-Jurado et al., 2018)<br>– Computationally prohibitive for extensive parametric analyses or repeated optimization cycles |
| **Surrogate-Based Hybrid Approach (Proposed)** | – Effectively balances computational effort and modeling accuracy<br>– Enables high-fidelity time-domain analyses at significantly reduced computational cost via surrogate modeling and design space reduction with analytical constraints<br>– Facilitates detailed structural optimization across multiple limit states (ULS, FLS, SLS)<br>– Provides a bridge between conceptual and detailed design phases<br>– Enhances accuracy significantly compared to purely frequency-domain approaches | – Requires cautious training and validation of surrogate models<br>– Performance is heavily dependent on the quality and representativeness of surrogate training data<br>– Initial computational burden associated with generating surrogate model training datasets |





Monte Carlo simulations for RBDO execution, while considering uncertainties (Leimeister and Kolios, 2021). Cousin et al. (2022) developed a two-step procedure for time-dependent RBDO problems and applied it to the design of a mooring line system for a semisubmersible floater. Initially, they reformulated the design constraints using ergodic theory and extreme value theory and then performed optimization using adaptive Kriging. More detailed reviews on FWT design optimization can be found in (Ojo et al., 2021; Patryniak et al., 2022; Zhang et al., 2022).

Pillai et al. (2018) compared the time domain and frequency domain methods for the mooring line geometry, observing that the frequency-domain model was underpredicting mooring line Damage Equivalent Loads (DELs). This underprediction led to selecting infeasible designs, which demonstrates that the usage of the frequency domain models is not suitable without using safety factors. The limitations of the frequency domain approach stem primarily from treating the dynamic systems as linear, resulting in a linear relationship between body kinematic variables. Additionally, without including an impulse response function for the hydrodynamics, it fails to account for how past movements affect the current behavior and ignores the memory effects (Journée and Massie, 2001).

Surrogate models have the ability to map complex relationships, and are frequently used to replace complex numerical models. This includes models relevant for FWTs and their design process, such as models for load time series simulation and prediction of systems properties. In Singh et al. (2025), prediction of 10-minute DELs is performed using a probabilistic surrogate model (Mixture Density Network) to compute conditional statistics while utilizing uncertainties due to site conditions. With this approach, they also minimized the training cost due to random seed repetitions. Other areas for surrogate modeling usage on FWTs can be the prediction of system properties. Baudino Bessone et al. (2024) applied a tree-based ensemble method as a surrogate modeling (XGBoost) technique for the Radiation diffraction analysis to predict the hydrodynamic coefficients with a mean error of 7%

The selection of the objective function for the optimization problem might also change the outcome of the process. The life cycle cost or the levelized cost of energy, which is subject to many unknowns and uncertainties, should be considered an objective function for the structural optimization of wind turbine structures (Muskulus and Schafhirt, 2014). This is especially important in defining the cost reduction potential of the design optimization methodology.

Additional structural optimization recommendations for the FWT systems can be stated as modeling with a hierarchy of fidelities to select suitable details for the selected stage, defining/reducing the design driving load cases, and exploring the probabilistic design possibilities while reducing the uncertainties in the system (Muskulus and Schafhirt, 2014).

This work introduces a novel two-step hybrid optimization framework specifically developed for FWT design, effectively addressing critical gaps in existing approaches (Table 1), and increasing the modelling fidelity achievable within automated design optimization of floating wind systems. The primary innovation lies in the strategic combination of analytical constraints and surrogate modeling techniques within a structured deterministic optimization procedure. Initially, analytical design constraints are systematically applied to substantially reduce the design space, efficiently excluding infeasible configurations. Subsequently, surrogate models—carefully trained using high-fidelity aero-hydro-elastic simulations—are employed to replace computationally intensive direct time-domain simulations within this refined design space.





The paper is organized as follows: Section 1 introduces the motivation, establishes the context, and reviews previous studies on FWT design optimization, highlighting the necessity and challenges associated with time-domain simulations. Section 2 details the methodology, describing the generation of the initial design space, parameterization of the FWT system, and the numerical modeling approach. Section 3 elaborates on the design optimization procedure, including the selection and definition of the design variables, the formulation of objective functions and constraints, and the overall optimization steps. Within

this, Section 3.5 specifically explains the surrogate modeling technique, detailing the dataset generation, sensitivity analysis to simulation seeds, and validation processes. Section 4 presents the optimization results, Section 5 provides a discussion interpreting these outcomes, and Section 6 offers concluding remarks and suggestions for future research directions.

## 2    Methodology

### 2.1    Overview

The methodology followed in this paper has four main elements: generation of input space, computation of system properties, surrogate modeling, and design optimization. This is illustrated in Figure 1. The approach presented here can be applied to different turbine and floater types.

The design process begins with the **design of experiment**, where a specific floater concept is selected for the optimization. At this stage, key design variables (denoted collectively as $\mathbf{X}_d$) and relevant environmental conditions, $\mathbf{X}_e$ are identified. Joint

site-specific distributions for $\mathbf{X}_e$, are defined, and samples of the design variables $\mathbf{X}_d$ are generated for the initial step of the design optimization. The augmented design vector is defined as $\mathbf{X} = [X_d, X_e]$.

Following this, the **system properties** for design configurations within the selected design space are computed. This involves evaluating initial design samples using analytical limit states to define the feasible design space. Once this space is identified, new samples of environmental variables $X_e$ and design variables $X_d$ are generated within the feasible region using

the predefined distributions.

The next phase focuses on **surrogate modeling**. A high-fidelity training database is created using an aero-elastic simulation tool, which serves as input for training surrogate models $\mathbf{S}_{Li}(X_e, X_d)$. This database is constructed for the normal operation of the wind turbine. Feedforward neural networks are trained to map environmental conditions and design variables for each quantity of interest (QoIs), including system responses, loads, and damage equivalent loads (DELs).

Finally, the last step is **design optimization** considering global limit states. In this phase, the surrogate models trained in the previous step are used to perform optimization. Samples are generated for the environmental variables $X_e$ for operational conditions, and the surrogate models are used to predict the QoIs for each limit state. The limit states computed are Ultimate Limit State (ULS), Serviceability Limit State (SLS), and Fatigue Limit State (FLS). For FLS, Monte Carlo simulations are performed as means of numerically integrating short-term fatigue damage estimates to obtain the lifetime fatigue damage.





**Figure 1.** Flowchart of the methodology





### 2.2 Description of the Reference FWT Design

The semisubmersible UMaine floater (Allen et al., 2020) is selected as the FWT reference design, due to its technical advantages, publicly available information, and commercialization preference for the semisubmersible concepts. The UMaine semisubmersible floater (see Figure 2 and Table 2) selected in this study is characterized by a steel substructure composed of three buoyancy columns connected by rectangular pontoons, and a central column supporting the wind turbine tower base. This configuration provides substantial hydrodynamic stability due to its relatively large waterplane area, which effectively limits platform motions, enhancing structural stiffness and robustness against wave-induced loading. Furthermore, the semisubmersible design provides efficient tow-out and installation processes due to its moderate draft requirements, reducing associated logistical and installation expenses. However, key disadvantages include its relatively complex structural design, potentially higher initial manufacturing and maintenance costs compared to simpler concepts such as spars, and vulnerability to fatigue due to significant wave interaction with the pontoons, which requires careful fatigue load assessment during design optimization. The reference design includes solid and water ballast. In this work, only solid ballast is preferred.

Semisubmersible platforms have increased waterplane area compared to spar platforms, contributing to better hydrodynamic stability, structural stiffness to resist wave loads, and greater towability, which ensures more straightforward installation and decommissioning properties (Jiang, 2021). Additional advantages include low draft requirements and lower mooring costs (Ojo et al., 2022).

**Table 2.** System Properties of the Floating Offshore Wind Turbine. Modified from (Allen et al., 2020)

| Parameter | Units | Value |
|---|---|---|
| Turbine Rating | MW | 15 |
| Hub Height | m | 150 |
| Excursion (L, W, H) | m | 90.1, 102.1, 290.0 |
| Freeboard | m | 15 |
| Draft | m | 20 |
| Total System Mass | t | 20,093 |
| Tower Mass | t | 1,263 |
| RNA Mass | t | 991 |
| Mooring System | – | Three-line chain catenary |

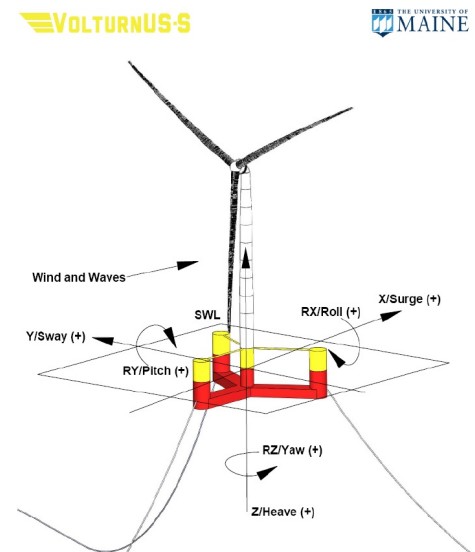

**Figure 2.** Selected reference design case (Allen et al., 2020)

The semisubmersible platform is coupled with an IEA 15 MW turbine (Gaertner, 2020) with a modified tower from the Hiperwind project (Capaldo et al., 2021b) and a controller with the tower top velocity feedback controller previously tuned





for the reference design. The controller with tower top fore-aft velocity feedback loop is designed to prevent the floater pitch instability problem (Meng et al., 2023). This method is developed as an alternative to detuning the controller to prevent pitch

instability and demonstrated by an experimental campaign for Tetra spar type FWT. For controller details, please see (Meng et al., 2023).

The reference design is adapted to a site in South Brittany at 150 m water depth with three equally spaced catenary mooring lines. The South Brittany site is selected due to having strong and consistent Atlantic winds. The water depth is also suitable for floating wind turbine installation while being relatively close to the shore. This site is also selected within the HIPERWIND

project (Capaldo et al., 2021a) for FWT analysis, and currently, there is an offshore wind farm planned in the region.

## 2.3    System Parameterization and Modeling

System parameterization is utilized in both steps of the optimization process. In the first step, the system properties, such as mass, stiffness, and natural frequencies, are computed for each design with respect to the current values of the design variables $\mathbf{X}_d$ (see Table 4 for an overview of all variables in $\mathbf{X}_e$ and $\mathbf{X}_d$). In the second step of the optimization simulation, additional

parameters are derived. For the hydrodynamic modeling, the selected reference floater design case is parameterized to generate the mesh used for computing the hydrodynamic coefficients of the floater. The mesh generation is automated using gmsh (Geuzaine and Remacle, 2009), considering the given design interval, and the mesh size is selected based on previous studies (See (Yildirim et al., 2024; Yildirim and Dimitrov, 2024) ). The Boundary Element Method solver HAMS / pyHAMS (Liu, 2019) calculates hydrodynamic coefficients. The equation of motion is solved in the time domain, and irregular waves are

generated using the JONSWAP spectrum according to the parameters selected in the DOE creation process. The wind turbine model of the IEA 15 MW turbine (Gaertner, 2020) is implemented in HAWC2. An overview of the tools used is presented in Table 3.

**Table 3.** Overview of tools for the framework

| Tool | Purpose/Function | Suitability |
| --- | --- | --- |
| **Gmsh** | Automated finite-element mesh generation for floater geometry. | Efficiently produces repeatable, high-quality meshes necessary for accurate hydrodynamic modeling in iterative design optimization. |
| **(py)HAMS** | Boundary Element Method solver for hydrodynamic coefficient calculation. | Precisely calculates hydrodynamic coefficients, essential for modeling floating structure dynamics. |
| HAWC2 | Aero-elastic simulation of the FWT dynamics. | Accurately simulates coupled aero-hydro-elastic dynamic responses, necessary for reliable assessment of structural loads, responses, and fatigue. |



## 2.4 Selection of the Design Variables

The contribution of the substructure cost to the total capital expenditure (CAPEX) of the FWT system is significant. For instance, a reference 6.1 MW turbine has around 27 % CAPEX contribution from substructure and foundation (Stehly and Duffy, 2021). Reducing the traditional factor of safety for the FWT systems can be one method to decrease this cost contribution. To ensure this, the floater is selected as the component for the design optimization study. Other system components, such as the tower, turbine, and station-keeping system, remain unchanged. Only the floater's design variables are changed during the optimization process.

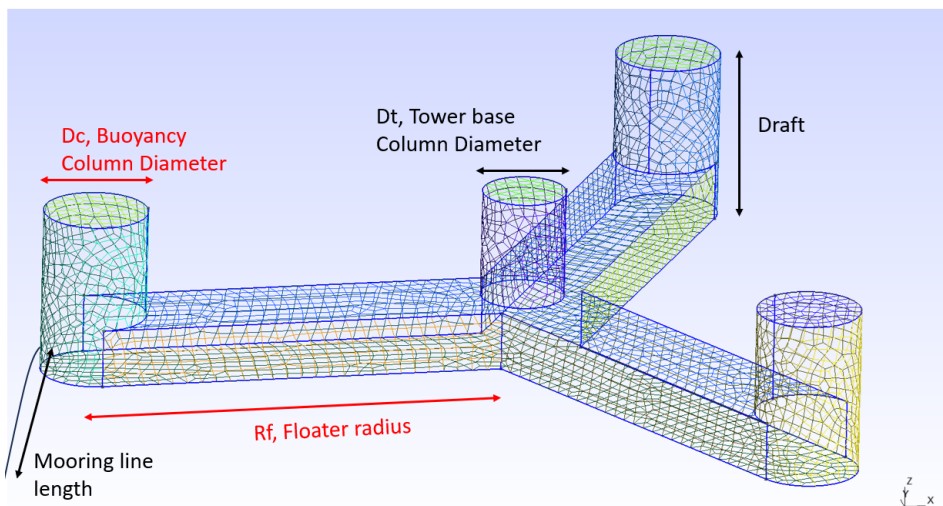

**Figure 3.** Visualization of selected design variables. Regenerated from (Yildirim et al., 2024).

The design variables considered in this study are defined based on a design evaluation and sensitivity analysis study for the floater design (Yildirim et al., 2024) conducted by the author. According to this study, floater radius and buoyancy column diameter are the design variables with the highest effect on the system response and the cost. Therefore, for the design optimization study, those two variables are selected for further design space exploration. Different from the reference design, the ballast of this system is taken as solid ballast material only and is adjusted based on the floater design and buoyancy of the total system.

To keep the same floater class (e.g., semisubmersible), the design interval is defined such that designs with very short and long drafts are eliminated from the initial design of experiment (DOE) screening. Finally, the design variables vector is defined as: $\mathbf{X_d} = \{x_1, x_2\}$ where $x_1$ is the buoyancy column diameter and $x_2$ is the floater radius. Selected design variables are presented in Figure 3 in red.

In this study, we focus specifically on two key design variables - floater radius and buoyancy column diameter - due to their substantial influence on system response and cost, as identified by prior sensitivity analyses. However, the optimization framework presented here is flexible and can readily accommodate additional variables in future studies. Variables such as





**Table 4.** Overview of the design and environmental variables

| Variable | Description |
| --- | --- |
| $U$ | 10 minutes averaged wind speed [ m/s ] |
| $Yaw_{mis}$ | Yaw misalignment [ ° ] |
| $\sigma_u$ | Standard deviation of the wind speed [ m/s ] |
| $H_s$ | Significant wave height [ m ] |
| $W_{dir}$ | Wave Direction [ ° ] |
| $T_p$ | Peak Wave Period [ s ] |
| $R_f$ | Flaoter Radius [ m ] |
| $D_{buoy}$ | Buoyancy Column Diameter [m] |

pontoon geometry, draft depth, ballast distribution, structural thickness, and mooring configuration can be integrated into the framework to facilitate more comprehensive design explorations. Such an extension would enable deeper insights and broader
optimization possibilities, particularly in detailed design phases where more complex interactions among multiple design parameters must be accurately captured and evaluated.

### 2.5 Definition of the Objective Function: LCOE

The optimization problem in this work is defined as a single-objective optimization problem. The objective function is formulated to minimize the levelized cost of energy (LCOE). By incorporating CAPEX, operational expenditures (OPEX), and
190 energy production over the system lifetime, LCOE enables a comprehensive assessment of design trade-offs, directly linking technical design decisions to economic outcomes. This approach facilitates transparent comparisons among competing FWT configurations and provides a robust basis for evaluating the practical impact of design optimizations. However, it should be acknowledged that uncertainties in cost estimation, market fluctuations, model uncertainties, and variable environmental conditions may affect the precision of LCOE calculations. Future studies on integrating probabilistic or uncertainty-based
assessments to further enhance decision-making robustness can increase the accuracy of the LCOE computations.

### 3 Design Optimization

The two steps in our proposed optimization approach make use of a common set of design variables $\mathbf{X}_d$ and environmental variables $\mathbf{X}_e$, as defined in Section 2. The remaining elements of the optimization problem include design constraints (analytical constraints evaluated in step 1, and dynamic limit state constraints used in step 2), an optimization algorithm, and an
200 objective function. All these are defined in the following.





## 3.1 Definition of the Design Constraints

The design constraints can be grouped in three types:

1) Analytical design constraints based on steady-state floater characteristics, motion and stability limits;

2) Design geometry constraints based on production and concept limits for the floater;

3) Global dynamic limit states (SLS, ULS, FLS as required by a standard design load basis) as well as maximum allowable motion limits

All constraints are defined as hard constraints, which refer to strict conditions that must be satisfied for a solution to be considered feasible. From the list above, the first two constraint groups are evaluated analytically during the first step of the optimization process, while the last group is computed with time domain simulations and surrogate modeling as part of step 2.

**Table 5.** Design Constraints for the First Stage of the Optimization

| Constraint | Expression | Description | Unit |
|---|---|---|---|
| | | **Analytical Design Constraints** | |
| $g_1(X)$ | $\omega_1 - 0.01$ | Surge Natural Frequency | Hz |
| $g_2(X)$ | $\omega_2 - 0.035$ | Pitch Natural Frequency (Lower Bound) | Hz |
| $g_3(X)$ | $0.025 - \omega_2$ | Pitch Natural Frequency (Upper Bound) (Pollini et al., 2023) | Hz |
| $g_4(X)$ | $\omega_3 - 0.469$ | Tower Natural Frequency | Hz |
| $g_5(X)$ | $-$system stability | System Stability (Pollini et al., 2023) | - |
| $g_6(X)$ | $\theta_{\text{static}} - 5$ | Static Pitch Angle (Pollini et al., 2023) | ° |
| | | **Constraints for the Design Variables** | |
| $g_7(x_1)$ | $25 - x_1$ | Floater Radius Lower Boundary | m |
| $g_8(x_1)$ | $x_1 - 80$ | Floater Radius Upper Boundary | m |
| $g_9(x_2)$ | $5.0 - x_2$ | Buoyancy Column Diameter Lower Boundary | m |
| $g_{10}(x_2)$ | $x_2 - 25.0$ | Buoyancy Column Diameter Upper Boundary | m |

The design constraints used in the first optimization stage are listed in Table 5. Analytical design constraints are given by $g_1(X)$ to $g_6(X)$, representing limits on natural frequencies, stability, and the static pitch angle. $g_7(x_1)$ and $g_8(x_1)$ represent lower and upper geometric boundaries for buoyancy column diameter $(x_1)$, while $g_9(x_2)$ and $g_{10}(x_2)$ are the floater radius $(x_2)$ lower and upper boundaries respectively.

## 3.2 Computation of Analytical Design Constraints

The first step of the design optimization process involves computing the system matrices, including mass and stiffness. The total stiffness matrix includes the contribution from the structure, the hydrostatic stiffness matrix, and the mooring stiffness matrices.





After computing the system matrices, the generalized eigenvalue problem is solved to obtain the system's natural frequencies. Mode partitioning is applied to differentiate contributions from each degree of freedom (DOF) to rank the different 'designs' natural frequencies. The system's stability is formulated using the total pitch stiffness of the structure. The results of the first design optimization part are presented in Figure 6 and Section 4.1 in detail.

### 3.3 Optimization Methodology

Optimization algorithms can be defined as two main categories: gradient-based and gradient-free algorithms. Gradient-based algorithms leverage derivative information to rapidly converge towards local optima but require smooth, differentiable functions, whereas gradient-free algorithms are more robust for complex, non-smooth, or discontinuous problems but typically converge more slowly and require more function evaluations.

The present work employs the Constrained Optimization by Quadratic Approximations (COBYQA) (Ragonneau, 2022) algorithm, which is a model-based derivative-free algorithm for nonlinear constrained optimization problems. COBYQA is a trust region approach and focuses on improving the local solution around the current iterate. COBYQA is particularly suitable for this study because it combines the strengths of gradient-free optimization with the efficiency and local accuracy typically associated with gradient-based methods. Specifically, it builds local quadratic approximations of the objective and constraint functions, enabling it to efficiently handle constrained nonlinear optimization problems without requiring explicit derivatives. This makes COBYQA especially effective for computationally expensive problems, such as surrogate-based design optimizations of floating wind turbines, where derivatives might be challenging to compute, yet accuracy, convergence reliability, and computational efficiency remain critical.

The details of the optimization algorithm can be found in (Ragonneau, 2022). COBYQA resulted in better convergence in our problem and had lower computational time compared to COBYLA, a similar, earlier version of the algorithm. A short overview of all optimization algorithms considered is given in Table 6. The generic optimization problem can be summarized as follows:

$$
\begin{aligned}
\text{Minimize:} \quad & f(\mathbf{x}) \\
\text{Subject to:} \quad & g_i(\mathbf{x}) \leq 0, \quad i = 1, 2, \ldots, m \\
& x_k^{\text{(lower)}} \leq x_k \leq x_k^{\text{(upper)}}, \quad k = 1, 2, \ldots, n
\end{aligned}
$$

Where $\mathbf{x} = [x_1, x_2, \ldots, x_n]^T$ is the vector of design variables, and for our case, there are two design variables and six environmental variables. For details, see Section 2.4. $f(\mathbf{x})$ is the objective function to be minimized, in our case represented by the LCOE (the computation of LCOE is described in the following Section 3.4). $g_i(\mathbf{x})$ are the inequality constraints and $x_k^{\text{(lower)}}$ and $x_k^{\text{(upper)}}$ are the lower and upper bounds for each variable. The optimization problem described in this work does not include any equality constraints.





**Table 6.** Gradient-Free Optimization Algorithms

| Algorithm | Description |
| --- | --- |
| COBYQA | Derivative-free trust-region method that constructs quadratic models of the objective and constraints to solve constrained nonlinear problems (Ragonneau, 2022). |
| COBYLA | Uses linear approximations to handle nonlinear constraints without requiring gradient information (pow, 1994). Can not handle variable boundaries directly. |
| Genetic Algorithms (GA) | Population-based metaheuristic inspired by natural selection; effective for exploring large, multi-modal, and nonlinear search spaces. (kus, 2012) |
| Particle Swarm Optimization (PSO) | Population-based algorithm, which is inspired by the social behavior of birds. (Kennedy' and Eberhart, 1995) It is effective for high-dimensional problems. |

## 3.4 Methodologies for Estimating LCOE under FWT Design Optimization

### 3.4.1 Computation of LCOE

LCOE is useful as an assessment tool for project viability as it provides a standard comparison by an economics-related metric. The LCOE by definition includes costs related to capital investments, operation, maintenance, and other project-related costs (Eq. 1). Estimation of LCOE for different FWT concepts, such as semisubmersible, tension leg platform (TLP), and spar, is compared in (Myhr et al., 2014), where they concluded that a floating wind array 100 km offshore has an LCOE range of 82 - 236.7 /MWh. They also identified the cost-driving aspects in two categories: discount rate, distance to shore, water depth,

and farm size are considered predictable factors, while load factor, variation of the steel price are the uncertain factors with high effects on the LCOE. In another study, different deployment sites and three floater concepts are considered for a 500 MW floating offshore wind farm where the final LCOE values are estimated between 67 - 135 (Lerch et al., 2018). They also highlighted that manufacturing-related costs highly influence the LCOE, such as the cost of the turbine, floater, and the station-keeping system, which states the importance of a cost-optimized design.

LCOE can be analyzed by different fidelities. This work prefers a simplified approach to compute the LCOE as an objective function. The LCOE is computed as defined in Equation 1.

$$\text{LCOE} = \frac{\sum_{t=1}^{T} \frac{\text{CAPEX}_t + \text{OPEX}_t}{(1+w)^t}}{\sum_{t=1}^{T} \frac{E_t}{(1+w)^t}} \tag{1}$$

Where $\text{CAPEX}_t$ is the capital expenditure in year $t$, $\text{OPEX}_t$ is the operational expenditure in year $t$, $E_t$ is the electricity generation in year $t$, $w$ is the discount rate which is 8 - 12 % for offshore wind projects (Myhr et al., 2014), $T$ is the lifetime

of the system. Here, the LCOE is computed based on the material mass of the system. Operational costs are included as a




percentage of the material cost. Due to early-stage floating wind markets, uncertain regulatory environments, and emerging FWT technology, the discount rate is selected as 12% to account for uncertainties and risks. For a typical offshore wind project, the lifetime of the structure is defined as 20-25 years. In this work, 25 years is selected as the lifetime of the project. For the computation of LCOE, the turbine price is taken from the (Agency, 2016). Details of the Annual Energy Production (AEP) computation are presented in the next section.

### 3.4.2 Computation of AEP

During the lifetime of the structure, the AEP is assumed to be constant; therefore, yearly wind resource variability is not considered. The probability distribution of the short-term (ten-minute) average wind speed is modelled as Weibull distributed, with the distribution parameters $A_w$ and $k_w$ (scale and shape parameters respectively) derived from the environmental data discussed in Section 3.5.1. The Weibull distribution is selected for wind speed modelling as it is widely adopted in practice when modelling wind energy potential (Ucar and Balo, 2010). The AEP computation formula is given in Equation 2:

$$\text{AEP} = \int_0^\infty F(u)P(u) \cdot du \tag{2}$$

where $u$ is the ten-minute average wind speed, $F(u)$ is the PDF of the Weibull distribution for the site, and $P(u)$ is the power curve computed for the reference FWT considered in this work. The AEP is calculated through detailed time-domain aero-elastic simulations performed with HAWC2, using inputs such as realistic turbulent wind fields, turbine control settings, and structural characteristics of the turbine and floater. These simulations yield power production and structural response outputs across operational wind speeds, which are subsequently integrated with site-specific wind distributions to estimate the annual energy yield. The annual energy production for a single turbine is calculated using Equation 2. AEP calculation is performed for two cases: 1) no turbulence and no wind shear, and 2) a turbulent case with IEC (International Electrotechnical Commission) NTM (Normal Turbulence Model) turbulence representative of the South Brittany Site. Six random seeds are considered for each case, to consider realization-to-realization variability due to turbulence as prescribed by the IEC 61400 design guideline (iec, 2019). The computed AEP values are 82.31 GWh for the no-turbulence case and 81.45 GWh for the IEC NTM turbulence model, which is 1.05 % lower than the former. This difference arises in the shoulder region of the power curve, where turbulence fluctuations at wind speeds close to rated will cause wind speed dips that lead to lower power production, but cannot be compensated with similar production peaks as the turbine hits is nominal power limit when the wind speed increases.

### 3.5 Surrogate Modeling Approach

The classical application of surrogate models is as a computationally efficient replacement of a high-fidelity tool in modelling a relationship between an input (design variable or environmental variable) and an output Quantity of Interest (QoI). This approach is applied in the present study in two ways. Firstly, wherever feasible, surrogate models are used to provide a direct mapping between design variables and output QoIs. This is applicable to the SLS and ULS limit states which are defined as single events. Computing the FLS limit state requires an additional step, where the lifetime fatigue damage is computed



from short-term DEL quantities through a numerical integration. In this case, the surrogate model is trained to estimate short-term DELs and the numerical integration is introduced as an additional step. The need of numerical integration increases the computation requirements and necessitates that a highly efficient surrogate model is chosen. In further consideration to the choice of surrogate, making gradient-based design optimization feasible requires at least the first-order differentials, a requirement which is satisfied by either polynomial-based models or a neural network with a continuous activation function (Dimitrov and Natarajan, 2021). In addition to this, gradient-enhanced surrogate modeling techniques can also increase the accuracy and computational efficiency of the optimization problem (Yamazaki et al., 2010).

Although neural networks may not be ideal for gradient-based optimization due to the tendency to overfit (which creates local extremes), they are very computationally efficient and have been found to be suitable for site-specific load estimation, considering time, accuracy, and convergence, particularly in a small sample space (Schröder et al., 2018). The requirement of high computational efficiency (in order to facilitate computation of FLS) and the availability of efficient gradient-free methods such as COBYQA, leads to the choice of feedforward neural networks (FNN) as surrogate models.

### 3.5.1 Generation of the Training Dataset

This paper benefits from training a site-specific surrogate model. The inputs include six environmental variables, including wind speed ($U$), wind direction ($U_{dir}$), standard deviation of the turbulence ($\sigma_u$), wave height ($H_s$), wave direction ($W_{dir}$), wave peak period ($T_p$) and two design variables as floater radius ($R_f$) and buoyancy column diameter ($D_{buoy}$). As the training dataset has significant importance for the quality of the surrogate model and optimization output, the design of experiment (DOE) generating the inputs is carefully defined to fill the input domain evenly.

A Latin Hypercube Sample (LHS) is generated over the space $X = [X_e, X_d]$. The space-filling properties of the LHS design typically result in a more accurate trained model (Zhang and Dimitrov, 2024). The joint distribution of environmental conditions is modelled as a series of conditionally dependent variables using the Rosenblatt transformation (Rosenblatt, 1952), thus ensuring the correlation between variables is properly accounted for. The boundaries for the design variables are defined considering the operation limits of the turbine and the floater stability class limitations. The environmental distributions for the surrogate model training are defined based on the site conditions in South Brittany (Vanem et al., 2023) using hourly data from the ANEMOC database containing 32 years of data (Forum, 2016). The six variable joint distributions previously defined are modified to represent the turbine's operational conditions. Instead of the hybrid Weibull and generalized Pareto distributions defined in (Vanem et al., 2023) for the wind speed, only a Weibull distribution is used as our interest is only in the operational wind speed range and resulted environmental condition pairplots are presented in Figure 4. The variable sequence in the joint conditional distribution is defined as in (Vanem et al., 2023):

$$f_{U,\sigma_U,HS,TP,\theta,\beta}(u,\sigma,h,t,\theta,\beta) =$$
$$f_U(u)f_{\sigma_U|U}(\sigma|u)f_{Hs|U}(h|u)$$
$$f_{Tp|Hs}(t|h)f_{\theta|U}(\theta|u)f_{\beta|U}(\beta|u) \tag{3}$$





It should be noted that the range given for the design variables here represents the initial design space before the surrogate training, and it is used for the second step of the optimization problem. For the $U$ values, the turbine operational range is preferred, and the limit states are also computed considering this operational range.

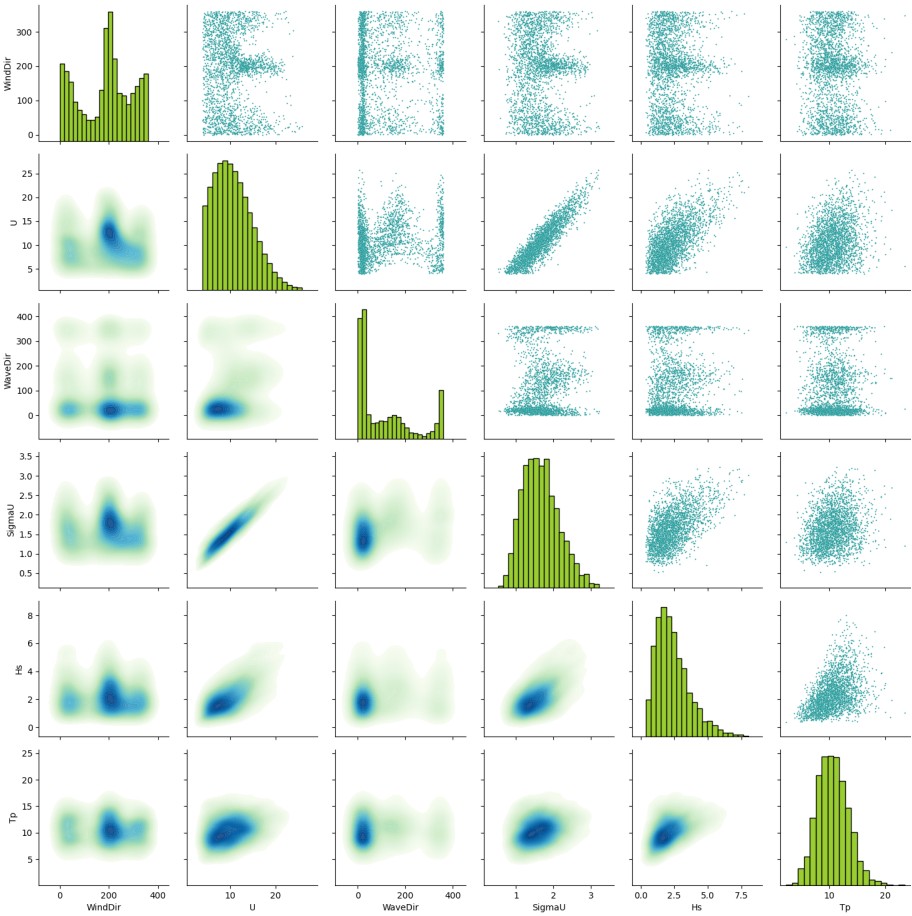

**Figure 4.** Environmental conditions for the normal turbulence model. Figure is regenerated from (Vanem et al., 2023) considering the changes in wind speed distribution.

For the surrogate model training, 3400 input sets are created based on the boundaries listed in Table 7. This number is selected as appropriate when considering the design space coverage and computational efficiency. The DOE space is defined as an eight-dimensional Latin hypercube in normalized space. After generating the sample space, the normalized samples are converted back to physical space, taking into account the boundaries discussed. Time domain simulations are performed with the Hawc2 tool (Horizontal Axis Wind Turbine Simulation Code 2nd Generation (Bischoff Kristiansen, 2022)), using

six random seeds per sample point, three for the wind turbulence and two for the waves. This results in a total of 20400 Hawc2 time-domain simulation data points. The resulting 20400 time series are post-processed to obtain the loads, responses,





and DELs on the structure. The 90% quantiles of the time series are used to further assess the ULS, to capture the high load/response of the system without being overconservative. The short-term DELs are computed for tower base, mooring lines, and blade roots by applying rainflow counting and the Palmgren-Miner's rule (iec, 2019) as in eq. 4 below, where the $S_i$ is the

340 load/stress amplitude of a number of cycles $n_i$ in the $i^{th}$ bin, and $m$ is the Wöhler exponent. The relationship between stress amplitudes and number of cycles to failure is derived from a standard SN curve where $N = QS^{-m}$. $n_{ref}$ is the number of equivalent cycles over a reference period (e.g., setting $n_{ref}$ to 600 for a 10-minute simulation results in 1 Hz-equivalent loads), and K is the total number of bins for DEL computation.

$$S_{eq} = \left[ \frac{\sum_{i=1}^{K} n_i S_i^m}{n_{ref}} \right]^{\frac{1}{m}} \tag{4}$$

$$DEL_{Lifetime} = \left[ \frac{N_{Lifetime}}{n_{ref} M} \sum^{M} S_{eq,i}^m \right]^{\frac{1}{m}} \tag{5}$$

Lifetime DEL is estimated using Equation 5, following the same notation as in Equation 4. Additional variables are defined as follows: $N_{Lifetime}$ is the total number of short-term periods over the lifetime of the system, $n_{ref}$ is the number of cycles for the 1 Hz equivalent simulation length and $M$ is the number of different environmental conditions used to compute $DEL_{Lifetime}$. $M$ is defined with a convergence study considering different numbers of environmental conditions, and it is selected as 50000

samples.

The DELs are estimated only for the operational range of the turbine. Note that Equation 5 holds when the lifetime DEL are computed by drawing a number of random samples equal to the total number of short-term periods corresponding to the operational life of the system (i.e., the computation effectively simulates the entire lifetime). This approach is convenient as it does not require probability weighting (each Monte Carlo sample is assumed equally likely), but it requires an efficient

surrogate model in order to be computationally feasible. The DEL computation makes use of an SN curve with an intercept of $Q = 6.0 \cdot 10^{10}$ and slope of $m = 3.0$ (Wöhler exponent), selected from the recommended values in DNV (2021) for studless chain mooring in corrosive environment. After estimating $DEL_{Lifetime}$, one can define the fatigue limit state for the mooring lines as below in Equation 6:

$$g_{Fatigue}(X) = \Delta - \frac{n_{eq} DEL_{Lifetime}^m}{Q} \tag{6}$$

where $\Delta$ is the fatigue damage limit for the material. In this work, the limit state is considered deterministic, and hence $\Delta$ is taken as 1.

A sensitivity analysis is conducted by considering six different cases for different seed configurations and four different simulation lengths to identify the required simulation length and number of seeds for a feasible DOE. Considered cases are presented in Table 9. Generally, 10-minute simulations are enough for representing turbulence characteristics, and it is im-

365 portant to separate mean wind conditions from turbulent fluctuations (Burton et al., 2011), but a longer simulation length is





**Table 7.** Input features for the surrogate model

| Variable | Distribution |
|----------|--------------|
| $U$ | Weibull |
| $U_{dir}$ | von Mises |
| $\sigma_u$ | Log Normal |
| $H_s$ | Weibull |
| $W_{dir}$ | von Mises |
| $T_p$ | Log Normal |
| $R_f$ | Uniform |
| $D_{buoy}$ | Uniform |

**Table 8.** Output features for the surrogate model

| Limit State | Quantity of Interest | Unit |
|-------------|----------------------|------|
| SLS | Surge | [m] |
| | Pitch | [°] |
| | Tower Top Acceleration fa | $[N/s^2]$ |
| | Tower Top Acceleration ss | $[N/s^2]$ |
| ULS | Pontoon Bending | [kN] |
| FLS | Mooring Line DEL | [Mpa] |

**Table 9.** Cases considered for the wind seed, wave seed, and simulation length.

| Case | Wind Seed | Wave Seed | Simulation Length [s] |
|------|-----------|-----------|------------------------|
| Case 1 | 3 | 1 | 800, 1400, 2000, 3800 |
| Case 2 | 3 | 2 | 800, 1400, 2000, 3800 |
| Case 3 | 4 | 2 | 800, 1400, 2000, 3800 |
| Case 4 | 4 | 3 | 800, 1400, 2000, 3800 |
| Case 5 | 5 | 3 | 800, 1400, 2000, 3800 |
| Case 6 | 6 | 3 | 800, 1400, 2000, 3800 |

vital to capture low-frequency dynamics/loads on the floating structure, which is essential for the floating wind turbines (due to lower surge natural frequencies, especially). For this sensitivity analysis, only a 12 m/s wind speed is considered, and the environmental conditions are selected from the distribution defined in Equation 3. As a measure of control, the median values of each simulation are used, and the percent root mean square (RMS) error is calculated based on six wind and three wave seeds. Considering the findings from this part, it is decided to use three wind seeds and two wave seeds with a 1400 s simulation length, including a 200 s transient period. The results are presented in Figue 5. This analysis is case dependent, and one should consult the relevant design guidelines such as (DNV, 2014; Veritas, 2010; iec, 2019).

### 3.5.2 Implementing the Surrogate Model

Surrogate model parameters (i.e., hyperparameters) should be tuned for the specific dataset, and there are several available methods for hyperparameter optimization. In this work, we opted to use Bayesian optimization. Bayesian optimization requires fewer iterations and converges to better optimal solutions in less time than traditional hyperparameter tuning algorithms, such as grid search and random search (Snoek et al., 2012). For the surrogate model fitting, we split the input dataset into test and training sets (20 % test and 80 % training), and used the test dataset to validate the model. The FNN architecture consists of





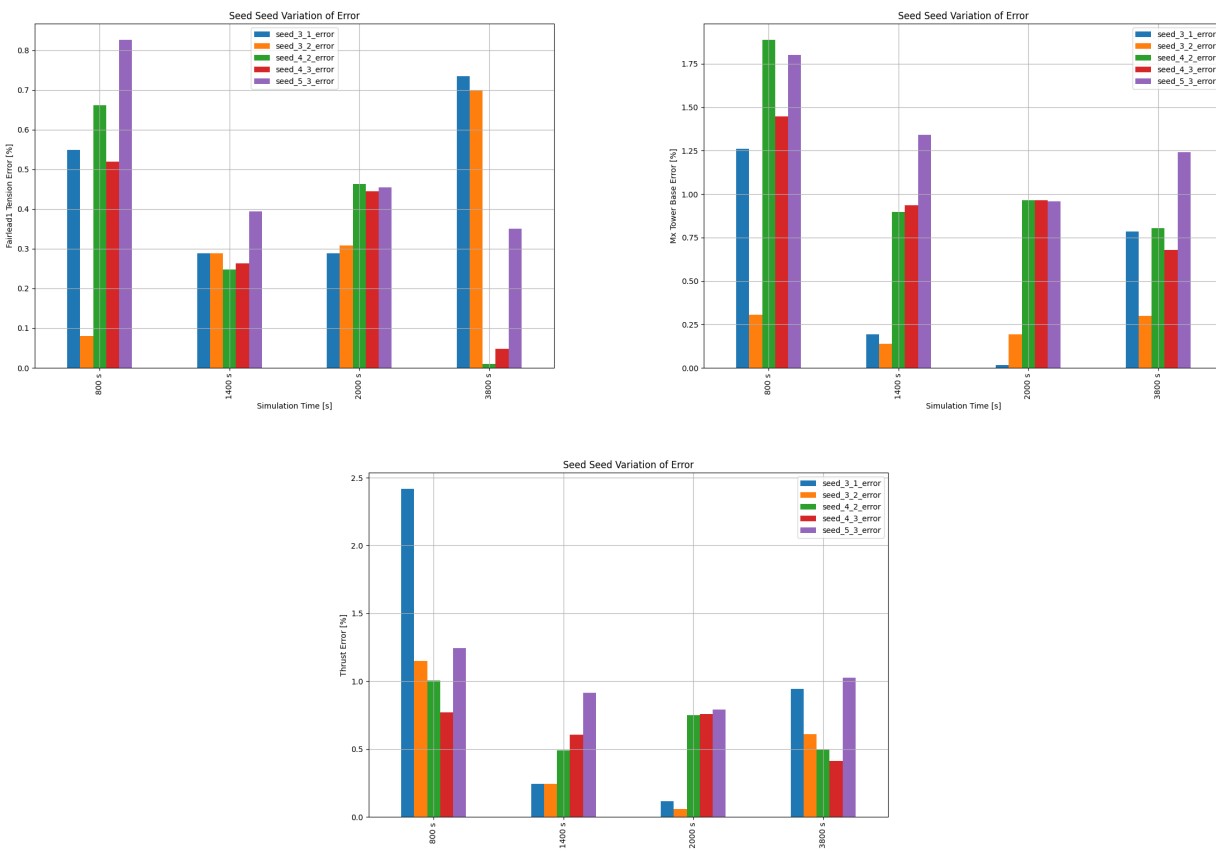

**Figure 5.** Percent Absolute Error for Different Number of Seeds

two hidden layers with Rectified Linear Unit (RELU) activation functions, and one output layer with a single neuron with a
linear activation function as required for regression tasks. Due to efficient gradient propagation, improved computational costs,
and good convergence properties (Alba et al., 2021), the RELU activation is preferred. The model accuracy is validated using
different error metrics, including RMSE and the coefficient of determination (R-squared value).

### 3.6 Second Step: Optimization Based on Global Structural Response Computed in Time Domain

Global system performance is preferred when investigating different floater characteristics and evaluating the limit states. The
global limit states considered in this study are divided into three main types: ultimate limit state (ULS), fatigue limit state
(FLS), and serviceability limit state (SLS). SLS is defined as the maximum allowable floater offset that avoids dynamic cable
damage; FOWT pitch angle, which determines system stability and power generation; and nacelle acceleration limit, which
prevents damage to the drivetrain and preserves its fatigue life. ULS is defined as a maximum mooring line tensile load, floater
pontoon buckling, and tower base buckling. FLS is defined in terms of the lifetime fatigue of the structure's mooring lines. The





design optimization in the time domain is conducted for multiple limit states simultaneously, while separate surrogate models are trained for each limit state based on the same training dataset. In Table 10, the global limit states for the second stage of the optimization are presented. SLS are given as $g_{11}(x_1)$ for dynamic surge motion, $g_{12}(X)$ for the dynamic pitch angle, $g_{13}(X)$ is for the maximum nacelle acceleration. $g_{14}(X)$ static pontoon bending $g_{15}(X)$ and maximum mooring line tension $g_{16}(X)$ are ULS and mooring line fatigue lifetime fatigue for mooring line 1 and mooring line 2 defined as $g_{16}(X)$ and $g_{17}(X)$.

**Table 10.** Design Constraints for the first part of the optimization

| Constraint | Description | Unit |
|:---:|:---|:---:|
| **Serviceability Limit States** | | |
| $g_{11}(X)$ | $\delta_1 - 40$ | m |
| $g_{12}(X)$ | $\delta_2 - 10$ (Dou et al., 2020; Leimeister et al., 2020b) | ° |
| $g_{13}(X)$ | $a_{nacelle} - 1.962$ (Leimeister et al., 2020b) | m/s$^2$ |
| **Ultimate Limit States** | | |
| $g_{14}(X)$ | $T_{max} - T_{cr}$ | kN |
| $g_{15}(X)$ | $\sigma_{design} - \sigma_{cr}$ | MPa |
| **Fatigue Limit States** | | |
| $g_{16}(X)$ | $D_{Life,moor1} - 1$ | - |
| $g_{17}(X)$ | $D_{Life,moor2} - 1$ | - |

# 4 Results

## 4.1 Low Cost Design Evaluation: First Step

This section presents the results of the analytical design constraint evaluation. The boundaries for each design constraint considered can be seen in Figure 6. When all analytical design constraints are considered, the final feasible design space is bounded by the static pitch angle and pitch natural frequency constraint. This result is concept and design variable-dependent, and different shapes can be obtained with different variables.

After running the analytical design constraint evaluation, the Pareto fronts of the feasible design space are defined to identify its boundaries. The exact boundary fit is offset to include potentially feasible design points that may not be captured by the relatively crude sampling used in the first optimization stage. We observe that in the present case the Pareto fronts can be approximated by a power law. The boundaries of a design space parameterized with a power law decay function can be seen in Figure 7a, and the resulting DOE for the time domain evaluation can be seen in Figure 7b.





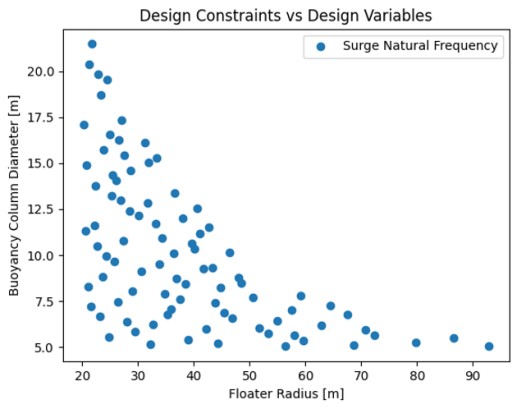

(a) Surge natural frequency constraint

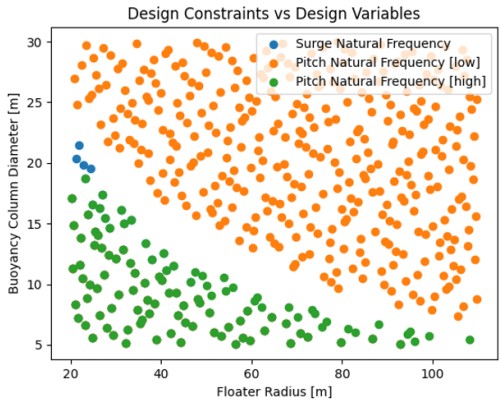

(c) Pitch natural frequency constraint (high)

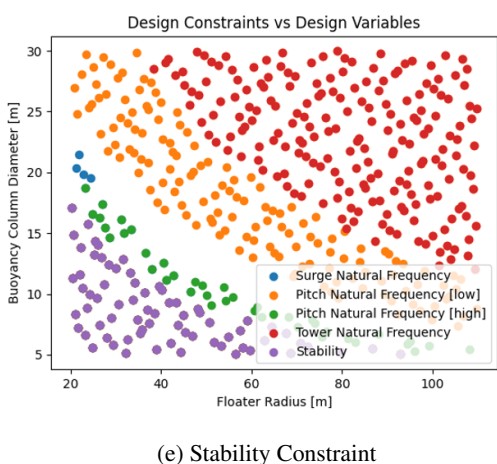

(e) Stability Constraint

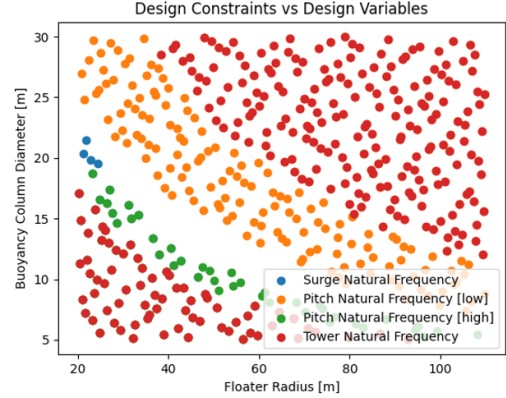

(b) Pitch natural frequency constraint (low)

(d) Tower natural frequency constraint

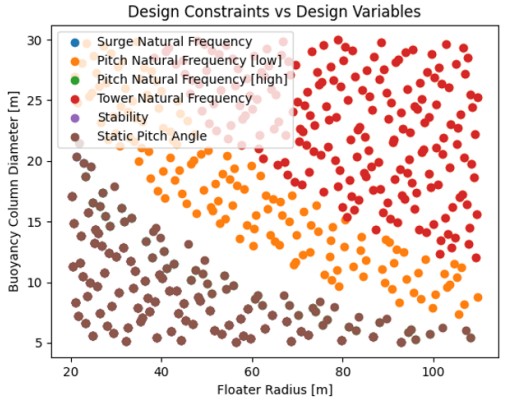

(f) Static Pitch Angle

**Figure 6.** Analytical design constraints evaluated sequentially for the initial DOE screening





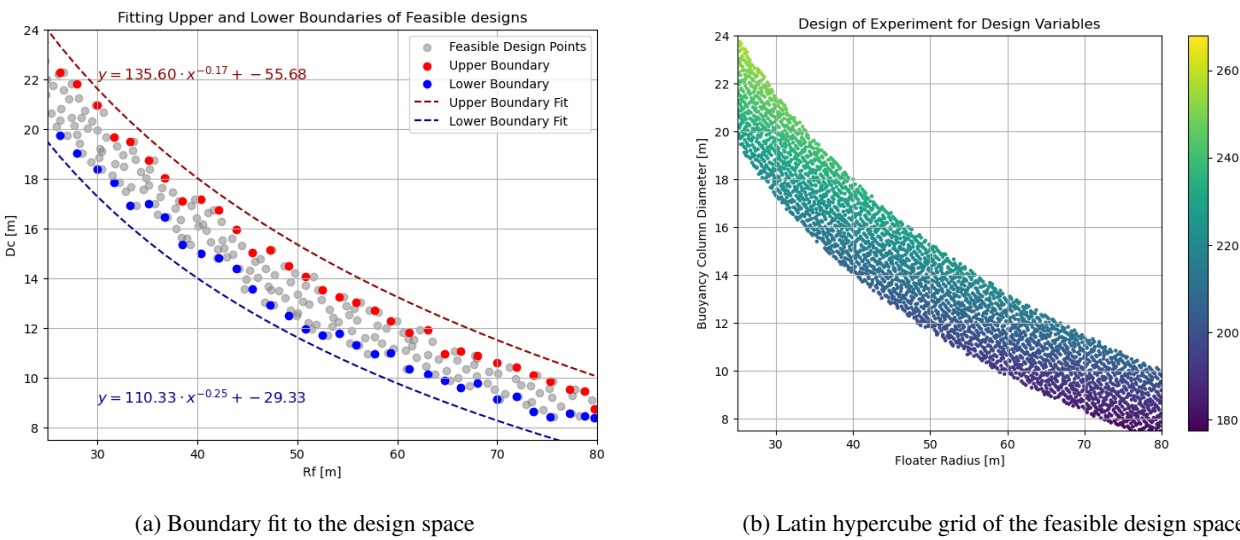

(a) Boundary fit to the design space        (b) Latin hypercube grid of the feasible design space

**Figure 7.** Refined design space after the analytical limit state evaluation

## 4.2 Time Domain Design Evaluation: Second Step

The second phase of the optimization includes results from the global limit states. Figure 8 presents the reference design and optimized design geometries side-by-side, and Figure 9 presents the iteration history of the design variables and objective function. As shown in Figure 10, the design driving constraints are mooring line fatigue and the lower boundary from the first step of the design optimization. The resulting design has 60.18 m floater radius and 10.3 m buoyancy column diameter with LCOE of 176.9 /MWh, which is a 3.7 % decrease from the nominal value.

## 5 Discussion

The method presented in this paper has several advantages, including the effective combination of analytical constraints and surrogate modeling to reduce computational complexity. It enables the evaluation of limit states based on high-fidelity time-domain analyses without excessive computational costs. Due to the flexibility of the framework, additional design variables or limit states can be added, which offers a practical balance between conceptual simplicity and detailed accuracy, ideal for iterative optimization processes.

The presented algorithm also faces limitations and challenges. A significant part of the computational cost associated with the use of the framework is due to the data generation required for surrogate model training. HPC simulations are currently the only feasible approach for generating the surrogate training set. In the present study, the simulations were performed on the Sophia HPC cluster owned by the Technical University of Denmark. Depending on the complexity of the aeroleastic model, this phase can be improved further. The computational cost for surrogate model training and for optimization (both the first





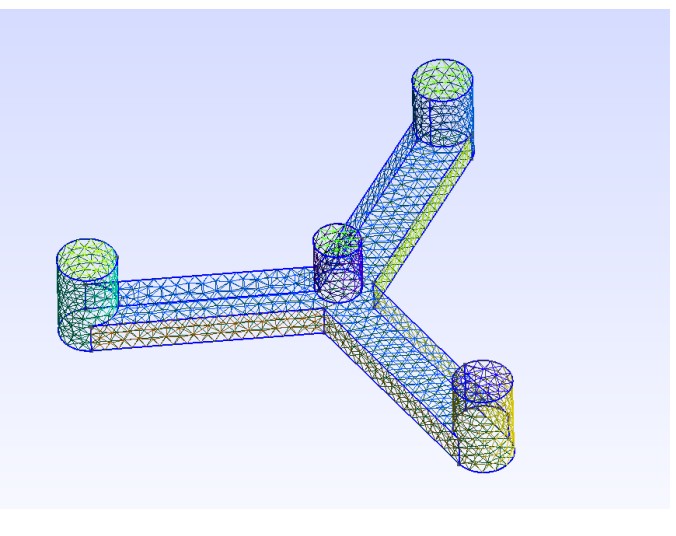

(a) Reference Design  (b) Optimized Design

**Figure 8.** Comparison of reference and optimized design

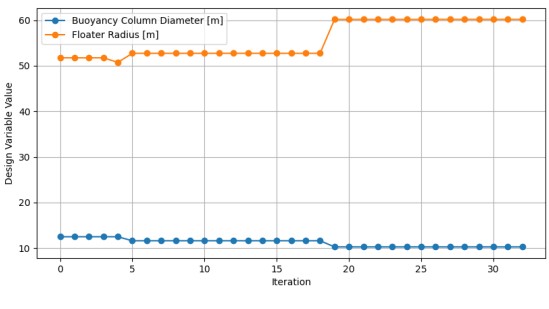

(a) Iteration history for the design variables  (b) Iteration history of the objective function

**Figure 9.** Design variables and cost values during the iteration for the optimization with mooring line FLS

step and the second step as described earlier) is in the order of minutes on a single CPU, which is negligible compared to the training data generation cost.

The cost of generating surrogate model training data has to be weighted against considerations on the accuracy of the resulting surrogate, as well as the representativeness of the design problem. Considering a limited number of design variables can cause neglect of additional influential design variables, such as structural thickness and mooring design properties.

The steel plate thickness is considered constant throughout the floater, which is likely not representative of the final floater design. This should be further investigated in the detailed design phase using FEM (Finite Element Modeling).

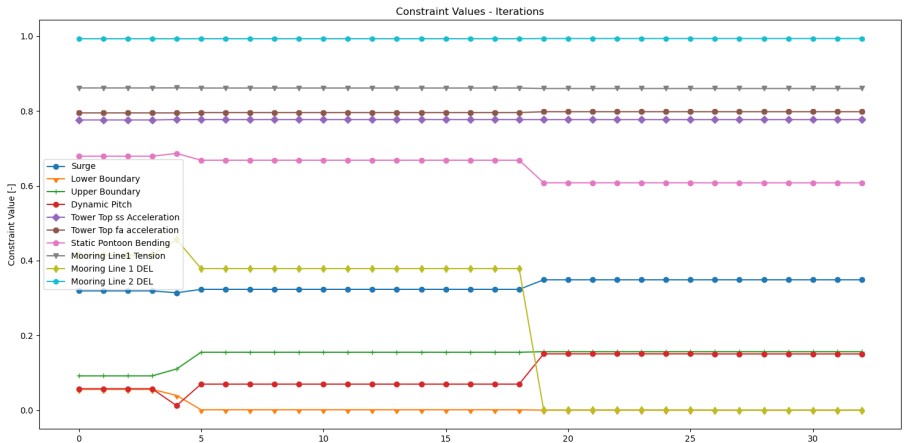

**Figure 10.** Constraint values for the iteration with mooring line FLS

The bending limit state is considered only for the static loading case. To indirectly include dynamic effects, an adjustment factor can be implemented within the optimization problem.

     Finally, the results obtained from this work demonstrated the potential of surrogate-based optimization methods for meaningful LCOE reduction in FWT designs. They also highlighted the feasibility and accuracy in bridging the gap between conceptual and detailed designs, which indicates significant potential for broader application across different FWT concepts and

larger-scale optimization problems. Future work should aim to expand the optimization scope to include more detailed design variables, incorporate robust or reliability-based design optimization to address uncertainties, and evaluate a wider range of optimization algorithms. Additionally, exploring probabilistic surrogate models could enhance the ability to quantify and manage uncertainties inherent in offshore wind conditions and cost estimations.

## 6   Conclusion

This study demonstrated a two-step surrogate-based optimization framework suitable to the design of floating wind turbine (FWT) substructures. The approach effectively implements analytical constraints to initially narrow down the feasible design space, and then applying surrogate models trained with high-fidelity time-domain simulations for detailed design evaluation against fatigue, serviceability, and ultimate limit states. By optimizing the buoyancy column diameter and floater radius of a UMaine semisubmersible platform coupled with the IEA 15 MW turbine, the proposed methodology achieved a meaningful

reduction of 3.7 % in the Levelized Cost of Energy (LCOE), ensuring that all global structural limit states - ultimate, fatigue, and serviceability - were met.

     The novelty of this work lies in the integration of analytical constraints and surrogate modeling within a deterministic design optimization procedure, enabling accurate, computationally efficient exploration of complex design spaces. This bridges the gap between conceptual simplicity and detailed accuracy, offering a practical optimization tool suitable for real-world FWT



design scenarios. Practically, the resulting framework not only facilitates significant economic improvements by reducing LCOE but also enhances design reliability, contributing directly to the advancement of floating offshore wind technology.

Future studies should expand this optimization framework by incorporating additional influential design variables, such as structural thickness, ballast configuration, and mooring system properties. Further research could integrate probabilistic surrogate models or robust/reliability-based design optimization methods to explicitly address the inherent uncertainties of
offshore wind environments. Lastly, evaluating alternative optimization algorithms and extending validation to other FWT concepts would enhance the robustness and applicability of this optimization approach across broader engineering contexts.

*Author contributions.*  BY: Conceptualization, methodology, investigation, writing (original draft), software, and visualization. ND: Conceptualization, methodology, writing (review and editing), and supervision. AK: Supervision, writing (review and editing). ABA: Supervision, methodology, writing (review and editing).

*Competing interests.*  The authors declare that one or more authors are members of the editorial board of WES journal.

*Acknowledgements.*  The authors gratefully acknowledge the computational and data resources provided on the Sophia HPC Cluster at the Technical University of Denmark, DOI: 10.57940/FAFC-6M81. This work is funded by the Department of Wind and Energy Systems at the Technical University of Denmark. The support is greatly appreciated.



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
