# Peer review of "Surrogate-Based Design Optimization of Floating Wind Turbines in Time Domain"

_Wind Energy Science, 2025_

## Author Comment (AC1)

Your reference: wes-2025-115 Date: October 14, 2025

DTU Wind and Energy Systems DTU Risø Campus Frederiksborgvej 399 4000, Roskilde Denmark

Subject: Author's Response Wind Energy Science Journal Reviewers

Dear Reviewers,

Thank you for your constructive and detailed comments on our paper. Your feedback helped us significantly improve our manuscript. This document presents our reply to the comments raised and provides an overview of the changes we made. Responses to each comment are given in blue color, and related changes in the manuscript are given in red color. A marked-up version of the revised manuscript is also attached, where the removed portions are indicated by red strikethrough and added text is indicated by blue underlined text. Hence, we provide a clear and detailed response to your comments.

Sincerely,

Büsra Yildirim Nikolay Dimitrov Asger Bech Abrahamsen Athanasios Kolios

**Enclosure(s):**

- General remarks
- Response to Reviewer 1
- Response to Reviewer 2
- Marked-up version of the revised manuscript

**Response to Reviewer 1**

The paper investigates the potential for optimizing floating wind turbines (FWTs) through a novel twostep hybrid optimization framework. This framework introduces innovation by combining design space reduction with surrogate-based modelling to enhance computational efficiency.

In the first step, the design space is reduced by applying design constraints aimed at excluding infeasible solutions. During this phase, the design variable vector  $X_d$  (representing buoyancy column dimensions and floater radius) and the environmental condition vector  $X_e$  are defined. Additionally, analytical design constraints are introduced based on the floater's dynamic behaviour and overall geometric limitations.

In the second step, a surrogate model based on feedforward neural networks is trained using aero-hydroservo-elastic simulations. This surrogate model enables efficient evaluation of the system's dynamic performance, significantly reducing computational cost. The design space and environmental conditions are further refined using Latin Hypercube Sampling.

The framework is demonstrated using the UMaine VolturnUS semisubmersible platform coupled with the IEA 15 MW Reference Wind Turbine (RWT). The optimization design variables include the external column diameter and floater radius. The objective is to minimize the Levelized Cost of Energy (LCOE), subject to constraints related to ultimate loads, serviceability, and fatigue performance.

**Comments and suggestions:**

1. The introductory part on the differences between design approaches is interesting and relevant to the study. It is suggested to further expand that section and also add additional references.

**Response:** Thank you for your comment. We expanded that section, including discussions about conceptual modeling [3, 2] and frequency domain optimization [6, 7].

**Revised Section:** Introduction is modified with additional references and discussion about conceptual design and frequency domain models.

2. The method is overall interesting. However, claiming that it is general for FOWT is too pretentious as this stage since there are design variables in the rotor (especially large and very flexible ones) and in the coupling between the rotor and the floater/moorings that still need to be proven as feasible with surrogate models. It is recommended that the paper title and abstract more clearly convey the fact that the procedure is so far applied only to floater design (and also assuming only some of the design variables).

**Response:** Thank you for your comment. We agree with your suggestion. We modified the abstract and title to present it more clearly that we optimized the floater only.

Revised Section: Title and abstract are modified.

3. The authors should better clarify the criteria with which the constraints in Table 5 have been selected.

**Response:** Thank you for your comment. We have added a more detailed explanation about the selection of the design constraints in Table 5, including references for tower natural frequency [7], stability [1], and static mean pitch angle [5]. The remaining constraints are also explained in more detail. In addition to Table 5, we have included more details on the design constraint for the second part of the optimization, utilizing references for dynamic surge motion [7] and nacelle acceleration [4].

**Revised Section:** We have added the discussion to Section 3.1. below Table 5.Additional discussion is added above Table 10 in Section 3.6.

4. To enhance the impact of the study, it is suggested that the authors give an estimation of the saving in terms of computational cost with respect to a direct optimization using the complete model in the time domain. In other words, the reader should understand better why it is important to have the two steps separately and use the surrogate model then: how long did this process take, with respect to using the complete model in HAWC2, run for several DLCs and seeds with different geometries? Are we getting the same accuracy? Is it worth it in terms of cost vs. accuracy?

**Response:** Thank you for your comment. In our approach, we reduced the design space based on the constraints we have and then built a surrogate model in the reduced space. With this approach,

we only reduced the design of the experiment for our surrogate model, which means that if we build the optimization in a single step and then apply the analytical constraints, those designs would be infeasible (all analytical constraints are geometry/structure dependent). Hence, we would waste computational time running simulations on the infeasible space. In terms of creating design space, this would result in a cost reduction of approximately  $89.4\,\%$ . If we compare the computational cost of using a surrogate model instead of aeroelastic simulations within the optimization, this would result in a computational reduction of  $98.72\,\%$ . Without the surrogate model, approximately  $1.6\,$  million simulations would need to be performed.

Revised Section: The discussion about the computational cost is included in the discussion.

5. A more critical discussion on the expected sensitivity of the results to the initial modelling choices, such as the selection of variables and constraints, would be valuable. Additionally, presenting the trends of key load responses in the optimized configuration compared to the baseline would help strengthen the analysis.

**Response:** Thank you for your comments. Additional simulations are performed with HAWC2 for the optimized design and the baseline case. We presented key load responses for the operational cases. Both designs show similar load trends.

**Revised Section:** Additional discussion is added to Section 4.2. Time Domain Design Evaluation for comparing key load responses of both designs.

**Response to Reviewer 2**

**A Major Comments**

1. The paper presents an interesting and promising optimization concept that aims to reduce the design space exploration of FOWTs for generating surrogate models to support design optimization. This is a valuable direction, and the approach has potential to be quite impactful.

**Response:** Thank you for your review and constructive comments. We are pleased to improve our paper using your suggestions and comments.

2. The most notable contribution seems to be the proposed two-step approach and the development of the surrogate model. At present, however, the details provided may not be sufficient for readers to fully understand or replicate the methodology. In particular, the rationale behind specific design choices and the interpretation of the optimization results could be expanded to improve clarity and accessibility.

**Response:** Thank you for your comments. To improve the clarity of our steps, we added additional details on design constraints for both steps, the design of the experiment (surrogate input) generation. We also extended the interpretation of optimization results.

**Revised Section:** We modified Section 3.1. Definition of Design Constraints, Section 3.5.1. Generation of the training dataset, Section 3.6. Second Step of the optimization, and Section 4. Results.

3. The computational cost of constructing the surrogate model is not fully described, apart from mentioning that a supercomputing cluster was required. More context here (e.g., runtime, resources, or scaling considerations) would help readers assess the feasibility of applying this approach in other settings. Relatedly, it would be helpful to know how the surrogate model itself (and the resulting optimal design) was verified.

Response: Thank you for your suggestions. We detailed the cost of surrogate training in terms of the reduction in the number of simulations for the optimization problem, rather than using the true model. More context about the computational cost is not included since the computational time also depends on the aeroelastic turbine simulation settings. For example, a simpler model with fewer structural nodes could be performed in a much shorter time. In our case, the whole database can be generated with 50 nodes within 1.5 days (Note that the nodes are not fully utilized for our simulations. It was used for other tasks on the cluster as well.). The parallelization of the time-domain simulation is not possible; therefore, each simulation runs independently. The surrogate model is verified using relevant error metrics, including R2 values. For the validation/comparison of

the optimized design, key load responses are provided through high-fidelity aeroelastic simulations. For further design verification, additional simulations should be performed, including other load cases.

**Revised Section:** Details on the cost of the surrogate model are included in Section 5, and discussion and key load responses are included in Section 4.2.

4. The description of the high-fidelity database could be clarified further. For instance, it is not entirely clear how environmental and design-based samples were combined. Since this step appears central to constructing a reliable surrogate, additional details would strengthen the paper and improve reproducibility.

Response: Thank you for your feedback. The design variables and environmental conditions are combined in a way that eliminates any correlation between them. By doing this, we aimed to ensure that we build our surrogate model without any dependency between environmental samples and design variables. To create the DOE, we initially perform the first step of the optimization, where, based on the analytical constraints, we obtain the feasible design area. We obtain equations for the boundaries, and then later we use them to filter samples we obtained from the uniform distribution and the initial range of the design variables. After this step, the same number of environmental conditions is generated, and then both are combined/concatenated to create DOE.

**Revised Section:** Discussion about the generation of DOE and combining design variables and environmental conditions are added to Section 3.5.1.

5. The discussion of LCOE uncertainties is thoughtful. This raises the question of whether structural mass might serve as a more robust figure of merit with less uncertainty. Additionally, it would be useful to clarify whether (and how) AEP changes with the platform model.

Response: Thank you for your response. In our approach to the design optimization, we considered that the AEP is constant for each design. The reason why we used LCOE as the objective of our optimization is that it ensures a better metric for comparison to the reference design. The power curve used for the AEP estimation is computed for the reference design (UMaine floater with IEA 15 MW turbine). A more detailed LCOE estimation could be more beneficial at the farm level design, which could be the next step in this optimization framework. Considering our assumptions, here using LCOE serves similar to using structural mass within the optimization. We also computed AEP for the optimized design (85.74 GWh) and reference design (85.93 GWh) where optimized design has 0.2211 % less AEP which can be caused by the larger pitch angle of the optimized system.

**Revised Section:** We added the key load/response comparison for the optimized and reference designs, including power generation comparison in Section 4.2. Additionally, the LCOE values are included for reference and optimized designs.

6. The sensitivity study could benefit from additional explanation. For example, how should the reader interpret the results shown in Figure 9? What is the reference for the reported errors?

Response: Thank you for your suggestion. Here, in Figure 9, we conducted a sensitivity study to examine the effects of varying the number of wind and wave seeds, as well as the simulation length. To perform this, we run simulations with different combinations and then compare the results with the case involving six wind and three wave seeds for each simulation length. Therefore, our reference is the case with the highest number of seeds (Six wind and three wave realizations). The results of this sensitivity analysis are presented for three quantities of interest, including mooring line tension, tower base moment, and the thrust force. The reason we selected these outputs is to represent the loads on different subsystems of the FWT system, which are subjected to various driving mechanisms. Here, the reader should interpret these results in light of the tradeoff between computational cost and accuracy. Statistically, using more realizations can increase the accuracy of postprocessing; however, we may also lose information on the floater's low-frequency response if the simulation length is not sufficiently long. So using the results in Figure 9, the readers can select the tradeoff between different numbers of wind and wave realizations for obtaining accurate statistics for a selected simulation length.

**Revised Section:** Figure 9 is modified and placed closer to where it is referred. The discussion is expanded in Section 3.5.1.

7. Figure 6 appears to play an important role in reducing the design space, but its meaning is somewhat difficult to interpret. For example, are the shown samples infeasible? How do the subfigures (a)–(f) relate to each other, and what do the different colors indicate? Adding clarification to the caption or text would improve readability.

**Response:** Thank you for your comment. In Figure 9, the shown samples are infeasible, and the different colors in the subfigures represent different design constraints and infeasible designs when they are computed. From Subfigures a to f, it is shown how we can decrease the feasible space with respect to different design constraints. This is useful for having more information about the design.

**Revised Section:** Section 4.1. is modified to include a more detailed discussion about Figure 9 and subfigures.

8. The process for building a Latin hypercube sampling from the feasible design space is not currently described, but seems to be an essential step in the study. Including this would strengthen the methodological transparency.

**Response:** Thank you for your comments. We agree that giving those details would strengthen the transparency and increase reproducibility. We included relevant discussion in Section 3.5.1. considering your previous comment (Major comment 4).

**Revised Section:** Discussion about the sampling from the feasible design space is described, and relevant text is added to Section 3.5.1.

9. The discussion of the final design optimization and results could be enriched. For example: which constraints are active? Which design variables changed, and why? What impact did these changes have on cost and constraints? Providing this interpretation would help highlight the significance of the results.

**Response:** Thank you for your comments. We enriched the discussion about optimization results.

**Revised Section:** We modified the discussion at Section 4.1. for the first step and at Section 4.2. for the final optimization results.

**B** Minor Comments**

1. The first sentence of the abstract highlights uncertainty, but this theme does not appear again later in the paper. A more consistent discussion might improve the narrative.

**Response:** Thank you for your suggestions. Currently, in our framework, we have addressed only environmental uncertainties, considering the full joint environmental distributions. Uncertainty propagation, including other uncertainties, can be included as further work. We modified the related section to include uncertainties in the discussion.

Revised Section: Discussion section is modified to include uncertainty related discussion.

2. The phrase "buoyancy column diameter" could be made clearer, as technically all columns provide buoyancy.

**Response:** Thank you for your suggestion. We replaced the "buoyancy column diameter" with the "outer column diameter".

**Revised Section:** Related text within the manuscript is modified.

3. Tables and figures are sometimes referenced far from where they appear in the text, which can disrupt the flow of reading.

**Response:** Thank you for your comment for improving flow. We improved the locations of the related figures. Figure 4 is moved closer to where it is referred.

**Revised Section:** Related figures are relocated within the text (Figure 4).

4. The definitions of design constraints could be presented more clearly. The  $g_i$  functions may not be necessary for readers, and plain language explanations might be more effective. Variable definitions should ideally appear in the captions if they are used within the tables (e.g., Tables 5 and 10). Additionally, the placement of these two tables feels quite far apart.

**Response:** Thank you for your suggestions. We added definitions and details on the design constraints. Additional details are discussed in Reviewer 1 Comment 3.

**Revised Section:** We have added the discussion to Section 3.1. below Table 5. Additional discussion is added above Table 10 in Section 3.6. Tables are relocated so that they are closer to where they are referred to. Captions for Tables 9 and 10 are modfied.

5. "System stability" appears in the constraints with only one citation and minimal explanation. More context would be helpful here.

**Response:** Thank you for your response. Additional details are added for system stability in the design constraints section.

Revised Section: Relevant text is added to Section 3.1 to give more details on the system stability.

6. The discussion of six random seeds is not entirely clear. Is this applied for each environmental case?

**Response:** Thank you for your comments. We gave more details about the six random seeds in your Major comment 6. In total, we have 3 wind and 2 wave seeds that are applied to each environmental case during the surrogate model input generation. Since our surrogate model is deterministic (each time we predict the same input with the same environmental conditions), we built our surrogate model based on the median of those random seeds for each environmental case.

Revised Section: We added more details in Section 3.5.1. for the random seeds discussion.

7. Table 6 lists several optimization algorithms, but it is not clear which one was actually used in the study. If only one is applied, it may be best to focus on that rather than listing all.

**Response:** Thank you for your comments. We removed Table 6. In our paper, we wanted to implement gradient-free algorithms, and that is why we summarized them and discussed why we selected COBYQA approach.

**Revised Section:** Table 6 is removed. More details are added about why gradient-free algorithms and why we selected COBYQA in Section 3.3.

8. For the DEL calculations, it would be valuable to explain the assumption that each sample is equally likely, as this may not be obvious to readers.

**Response:** Thank you for your suggestion. We modified the discussion about the DEL calculations and explained that our sample space is large enough to consider full joint probability distributions of each sample, and we don't need to scale them. Even if we increase our selected M value, the result for  $DEL_{Lifetime}$  wouldn't change.

**Revised Section:** Section 3.5.1. is modified to explain our assumption about equally likely environmental conditions.

9. The distributions of the environmental parameters are not described. Were they fitted to metocean data? Including this information would be helpful.

**Response:** Thank you for your suggestion. The environmental parameters are fitted to the reanalysis data from the ANEMOC database within the HIPERWIND project [8]. The joint distribution covers 32 years of data. The underlying distributions for each environmental condition are presented in Table 6.

**Revised Section:** For better interpretation, the related section is modified further in Section 3.5.1 (Generation of the Dataset)

10. Section 3.6: the sentence beginning "SLS is defined as the maximum..." is difficult to parse and could be revised for clarity.

**Response:** Thank you for your suggestions. We agree with your suggestions and paraphrased the sentence.

**Revised Section:** We paraphrased the sentence beginning "SLS is defined as the maximum..." in Section 3.6. for clarity.

**C Style Comments**

1. Both "FOWT" and "FWT" are used; standardizing terminology would improve consistency.

**Response:** We agree about this inconsistency. The initial idea was to use FWT as the standard terminology.

Revised Section: "FOWTs" within the text are changed to "FWT".

2. Sideways tables can be challenging to read. If possible, reformatting them would enhance readability.

Response: Thank you for this suggestion. We reformatted the table for better readability.

**Revised Section:** Table 1 is reformatted for enhanced readability.

3. Figure labels should be consistent with the text size. At present, Figures 4–9 are difficult to read.

**Response:** Thank you for your suggestions. To improve the readability of our manuscript, we modified Figures 4-9.

**Revised Section:** Figures 4 - 9 are modified to improve consistency and readability.

**References**

- [1] A. Biran and R. López-Pulido. *Ship hydrostatics and stability*. Elsevier: Butterworth-Heinemann, Amsterdam, second edition edition, 2014.
- [2] M. Borg and H. Bredmose. D4.4 Overview of the numerical models used in the consortium and their qualification. 2015.
- [3] F. Borisade, J. Gruber, L. Hagemann, M. Kretschmer, F. Lemmer, K. Müller, D. Schliph, N.-D. Nguyen, and L. Vita. D 7.4 State-of-the-Art FOWT design practice and guidelines. Unrestricted, University of Stuttgart, Stuttgart, 2016.
- [4] M. Leimeister, A. Kolios, M. Collu, and P. Thomas. Design optimization of the OC3 phase IV floating spar-buoy, based on global limit states. *Ocean Engineering*, 202:107186, Apr. 2020.
- [5] D. Matha, F. Sandner, C. Molins, A. Campos, and P. W. Cheng. Efficient preliminary floating offshore wind turbine design and testing methodologies and application to a concrete spar design. *Philosophical Transactions of the Royal Society A: Mathematical, Physical and Engineering Sciences*, 373(2035):20140350, Feb. 2015.
- [6] A. Pegalajar-Jurado, M. Borg, and H. Bredmose. An efficient frequency-domain model for quick load analysis of floating offshore wind turbines. *Wind Energy Science*, 3(2):693–712, Oct. 2018.
- [7] N. Pollini, A. Pegalajar-Jurado, and H. Bredmose. Design optimization of a TetraSpar-type floater and tower for the IEA Wind 15 MW reference wind turbine. *Marine Structures*, 90:103437, July 2023.
- [8] E. Vanem, E. Fekhari, N. Dimitrov, M. Kelly, A. Cousin, and M. Guiton. A Joint Probability Distribution Model for Multivariate Wind and Wave Conditions. In *Volume 2: Structures, Safety, and Reliability*, page V002T02A013, Melbourne, Australia, June 2023. American Society of Mechanical Engineers.

**Surrogate-Based Design Optimization of Floating-Wind Turbines Turbine Floater in Time Domain**

Büsra Yildirim1, Nikolay Dimitrov1, Athanasios Kolios1, and Asger Bech Abrahamsen1 1DTU Wind and Energy Systems, Frederiksborgvej 399, DK-4000, Roskilde, Denmark

Correspondence: Büsra Yildirim (bysyi@dtu.dk)

**Abstract.**

Floating wind turbine (FWT) design involves higher costs and greater uncertainty than onshore or fixed-bottom offshore turbines due to low technology maturity, limited operational experience, and harsh marine environments; these factors have led to conservative design practices. To address these challenges, we introduce a novel two-step deterministic surrogate-based optimization framework that enables efficient time-domain design optimization for the floater subsystem of FWTs. In the first step, analytical design constraints are applied to refine the design space and establish a feasible region. In the second step, a surrogate model is trained on high-fidelity aero-hydro-elastic simulations, covering the reduced design space defined from step 1. During an optimization run, the surrogate model replaces computationally expensive direct time-domain analyses, capturing the dynamic response of the system with significantly reduced computational effort. This approach effectively balances model fidelity and computational cost, bridging the gap between conceptual and detailed design phases for floating wind structures. We demonstrate the framework on a semisubmersible platform floater (UMaine VolturnUS) coupled with the IEA 15 MW reference wind turbine, a representative large-scale FWT. Two primary design variables — the buoyancy of the floater, the outer column diameter and the overall floater radius, are optimized to minimize the levelized cost of energy (LCOE) of the system. The optimization incorporates global structural limit state constraints covering ultimate (ULS), fatigue (FLS), and serviceability (SLS) requirements to ensure the design's structural feasibility. The surrogate-assisted optimization yields a design that achieves a LCOE of 176.9 €/MWh, which is a 3.7 % reduction in LCOE relative to the baseline, with feasibility validated against all ULS, FLS, and SLS criteria. These results highlight the framework's potential to reduce FWT costs and improve design reliability by enabling time-domain optimization without excessive computational expense.

**1 Introduction**

[revised manuscript text omitted]

| Modeling                                                                                                                                                                                                                                                  | Approach                                                                                                                                 |
|-----------------------------------------------------------------------------------------------------------------------------------------------------------------------------------------------------------------------------------------------------------|------------------------------------------------------------------------------------------------------------------------------------------|
| Advantages                                                                                                                                                                                                                                                | Disadvantages                                                                                                                            |
| Frequency-Domain Models                                                                                                                                                                                                                                   | Frequency-Domain Models                                                                                                                  |
| Computationally efficient (fast calculations) (Borg and Collu, 2015) Suitable for preliminary and conceptual design phases  Enables rapid exploration of large design spaces Facilitates  multi-objective optimization with reduced computational demand- | Assumes linear system behavior, limiting accuracy (Borg and Collu, 2015; Journée and Massie, 2001)                                       |
| Suitable for preliminary and conceptual design phases                                                                                                                                                                                                     | Underpredicts dynamic and nonlinear structural responses                                                                                 |
| Enables rapid exploration of large design spaces                                                                                                                                                                                                          | 
[revised manuscript text omitted]
_{\sim}$                         | 10 minutes averaged wind speed [ m/s ]       |
| $\underbrace{Yaw_{mis}}$           | Yaw misalignment [ ° ]                       |
| $\overset{\sigma_u}{\sim}$         | Standard deviation of the wind speed [ m/s ] |
| $\stackrel{H_s}{\approx}$          | Significant wave height [m]                  |
| $W_{dir}$                          | Wave Direction [ ° ]                         |
| $T_{\mathcal{R}}$                  | Peak Wave Period [ s ]                       |
| $\stackrel{R_{f_{\sim}}}{\approx}$ | Flaoter Radius [ m ]                         |
| $D_{buou}$                         | Outer Column Diameter [m]                    |

**Table 4.** Overview of tools for the framework

| Tool     | Purpose/Function                 | Suitability                                            |  |
|----------|----------------------------------|--------------------------------------------------------|--|
| Gmsh     | Automated finite-element mesh    | Efficiently produces repeatable, high-quality meshes   |  |
|          | generation for floater geometry. | necessary for accurate hydrodynamic modeling in iter-  |  |
|          |                                  | ative design optimization.                             |  |
| (py)HAMS | Boundary Element Method          | Precisely calculates hydrodynamic coefficients, essen- |  |
|          | solver for hydrodynamic coef-    | tial for modeling floating structure dynamics.         |  |
|          | ficient calculation.             |                                                        |  |
| HAWC2    | Aero-elastic simulation of the   | Accurately simulates coupled aero-hydro-elastic dy-    |  |
|          | FWT dynamics.                    | namic responses, necessary for reliable assessment of  |  |
|          |                                  | structural loads, responses, and fatigue.              |  |
| -        |                                  |                                                        |  |

the tower, turbine, and station-keeping system, remain unchanged. Only the floater's design variables are changed during the optimization process.

The design variables considered in this study are defined based on a design evaluation and sensitivity analysis study for the floater design (Yildirim et al., 2024) conducted by the author. According to this study, floater radius and buoyancy outer column diameter are the design variables with the highest effect on the system response and the cost compared to other variables, such as tower base column diameter, draft, and the mooring line length. Regarding cost comparison, the outer column diameter has

[revised manuscript text omitted]

The design constraints used in the first optimization stage are listed in Table 5. Analytical design constraints are given specified by  $g_1(X)$  to  $g_6(X)$ , representing which represent limits on natural frequencies, stability, and the static pitch angle under rated wind speed.  $g_7(x_1)$  and  $g_8(x_1)$  represent lower and upper geometric boundaries for buoyancy outer column diameter  $(x_1)$ , while  $g_9(x_2)$  and  $g_{10}(x_2)$  are the floater radius  $(x_2)$  lower and upper boundaries, respectively.

Design constraints for the first part of the optimization are selected to eliminate infeasible designs before running simulations, thereby reducing computational cost. The natural frequency constraints, as surge  $(g_1(X))$  and pitch  $(g_2(X))$  and  $g_3(X)$ , are selected to prevent resonance due to load frequencies resulting from wave loads and controller behavior. For the tower,

**Table 5.** Design Constraints for the First Stage of the Optimization where  $\omega_1$  is the surge natural frequency,  $\omega_2$  is the pitch natural frequency,  $\omega_3$  is the tower natural frequency of the system.  $\theta_{\text{static}}$  is the static pitch angle under thrust force at rated wind speed.  $x_1$  and  $x_2$  are our design variables where  $x_1$  is the floater radius length and  $x_2$  is the outer column diameter.

| Constraint    | Expression                           | Description                                                  | Unit |  |  |  |
|---------------|--------------------------------------|--------------------------------------------------------------|------|--|--|--|
|               | Analytical Design Constraints        |                                                              |      |  |  |  |
| $g_1(X)$      | $\omega_1 - 0.01$                    | Surge Natural Frequency                                      | Hz   |  |  |  |
| $g_2(X)$      | $\omega_2 - 0.035$                   | Pitch Natural Frequency (Lower Bound)                        | Hz   |  |  |  |
| $g_3(X)$      | $0.025 - \omega_2$                   | Pitch Natural Frequency (Upper Bound) (Pollini et al., 2023) | Hz   |  |  |  |
| $g_4(X)$      | $\omega_3 - 0.469$                   | Tower Natural Frequency                                      | Hz   |  |  |  |
| $g_5(X)$      | -system stability                    | System Stability (Pollini et al., 2023)                      | -    |  |  |  |
| $g_6(X)$      | $\theta_{ m static} - 5$             | Static Pitch Angle (Pollini et al., 2023)                    | 0    |  |  |  |
|               | Constraints for the Design Variables |                                                              |      |  |  |  |
| $g_7(x_1)$    | $25 - x_1$                           | Floater Radius Lower Boundary                                | m    |  |  |  |
| $g_8(x_1)$    | $x_1 - 80$                           | Floater Radius Upper Boundary                                | m    |  |  |  |
| $g_9(x_2)$    | $5.0 - x_2$                          | Buoyancy Outer Column Diameter Lower Boundary                | m    |  |  |  |
| $g_{10}(x_2)$ | $x_2 - 25.0$                         | Buoyancy Outer Column Diameter Upper Boundary                | m    |  |  |  |

the natural frequency constraint is selected for stiff-stiff tower design where the tower's natural frequency should be placed above the 3P region (Pollini et al., 2023). The hydrostatic pitch stability of a floating body can be expressed in terms of the hydrostatic pitch restoring stiffness, and it should be positive for the floater to be stable. This stability constraint can be explained by the stability analysis of floating bodies, which considers the center of gravity, buoyancy center, and metacenter (Biran and López-Pulido, 2014). Static mean pitch angle is constrained with a strict 5 °criteria at rated wind speed to ensure stability and efficient power production (Matha et al., 2015).

**3.2 Computation of Analytical Design Constraints**

235

245

The first step of the design optimization process involves computing the system matrices, including mass and stiffness. The total stiffness matrix includes the contribution from the structure, the hydrostatic stiffness matrix, and the mooring stiffness matrices.

After computing the system matrices, the generalized eigenvalue problem is solved to obtain the system's natural frequencies. Mode partitioning is applied to differentiate contributions from each degree of freedom (DOF) to rank the different 'designs' natural frequencies. The system's stability is formulated using the total pitch stiffness of the structure. The results of the first design optimization part are presented in Figure 6 and Section 4.1 in detail.

**3.3 Optimization Methodology**

250

255

260

265

270

Optimization algorithms can be defined as two main categories: gradient-based and gradient-free algorithms. Gradient-based algorithms leverage derivative information to rapidly converge towards local optima but require smooth, differentiable functions, whereas gradient-free algorithms are more robust for complex, non-smooth, or discontinuous problems but typically converge more slowly and require more function evaluations.

The present work employs the Some of the gradient-free optimization algorithms are inspired by nature. For example, the genetic algorithm (GA) is a population-based metaheuristic algorithm that is effective for exploring large, multimodal, and nonlinear search spaces (kus, 2012). Particle swarm optimization (PSO), which is inspired by the social behavior of birds Kennedy' and Eberhart (1995), is effective for high-dimensional problems. Constrained Optimization by Linear Approximations (COBYLA) uses linear approximations to handle nonlinear constraints without requiring gradient information (pow, 1994) and can not handle variable boundaries directly. Constrained Optimization by Quadratic Approximations (COBYQA) (Ragonneau, 2022) is an improved version of the COBYLA algorithm, which constructs quadratic models of the objective and constraints to solve constrained nonlinear problems (Ragonneau, 2022).

The present work employs the COBYQA (Ragonneau, 2022) algorithm, which is a model-based derivative-free algorithm for nonlinear constrained optimization problems. COBYQA is a trust region approach and focuses on improving the local solution around the current iterate. COBYQA is particularly suitable for this study because it combines the strengths of gradient-free optimization with the efficiency and local accuracy typically associated with gradient-based methods. Specifically, it builds local quadratic approximations of the objective and constraint functions, enabling it to efficiently handle constrained nonlinear optimization problems without requiring explicit derivatives. This makes COBYQA especially effective for computationally expensive problems, such as surrogate-based design optimizations of floating wind turbines, where derivatives might be challenging to compute, yet accuracy, convergence reliability, and computational efficiency remain critical.

The details of the optimization algorithm can be found in (Ragonneau, 2022). COBYQA resulted in better convergence in our problem and had lower computational time compared to COBYLA, a similar, earlier version of the algorithm. A short overview of all optimization algorithms considered is given in Table ??. The generic optimization problem can be summarized as follows:

Minimize:
$$f(\mathbf{x})$$
 Subject to:  $g_i(\mathbf{x}) \leq 0, \quad i=1,2,\ldots,m$
$$x_k^{(\mathrm{lower})} \leq x_k \leq x_k^{(\mathrm{upper})}, \quad k=1,2,\ldots,n$$

[revised manuscript text omitted]
 Vanem et al. (2023) fitted a multivariate joint distribution to the site conditions, where the wind speed distribution is modeled as a hybrid model of Weibull - Generalized Pareto Distribution (GPD) for a better representation of extremes. This six-variable joint distribution is adapted in our work with focus on the range of the turbine's operational conditions. Instead of the hybrid Weibull and generalized Pareto distributions Weibull-GPD defined in (Vanem et al., 2023) for the wind speed, only a Weibull distribution is used, as our interest is only in the operational wind speed rangeand resulted, and the resulting environmental condition pairplots are presented in Figure 4. The variable sequence in the joint conditional distribution is defined as in (Vanem et al., 2023):

$$f_{U,\sigma_{U},HS,TP,\theta,\beta}(u,\sigma,h,t,\theta,\beta) = f_{U}(u)f_{\sigma_{U}|U}(\sigma|u)f_{Hs|U}(h|u)f_{Tp|Hs}(t|h)f_{\theta|U}(\theta|u)f_{\beta|U}(\beta|u)$$

$$(3)$$

It should be noted that the range given for the design variables here represents the initial design space before the surrogate training, and it is used for the second step of the optimization problem. For the *U* values, the turbine operational range is preferred, and the limit states are also computed considering this operational range.

**Figure 4.** Environmental conditions for the normal turbulence model. Figure is regenerated from (Vanem et al., 2023) considering the changes in wind speed distribution.

For the surrogate model training, 3400 input sets are created based on the boundaries distributions listed in Table 6. This number is selected as appropriate when considering the design space coverage and computational efficiency. The DOE space is defined as an eight-dimensional Latin hypercube in normalized space. After generating the sample space, the normalized samples are converted back to physical space, taking into account the boundaries discussed. During sample generation for the DOE, environmental distributions and design variables were considered separately to ensure that design variables and environmental conditions are not dependent on each other. For sampling the design variables, initially, uniform distributions are defined based on the design variable boundaries. After performing the first part of the design optimization, feasible design

375

variable boundaries are fitted by binning the variable range and then filtered to obtain a feasible space. The same number of environmental conditions are sampled and then combined.

**Table 6.** Input features for the surrogate model

380

385

390

Table 7. Output features for the surrogate model

|                            |                                                                                                                                                                                                                                                                                                                                                                                                                                                                                                                                                                                                                                                                                                                                                                                                                                                                                                                                                                                                                                                                                                                                                                                                                                                                                                                                                                                                                                                                                                                                                                                                                                                                                                                                                                                                                                                                                                                                                                                                                                                                                                                               |    |          |                           |                  |
|----------------------------|-------------------------------------------------------------------------------------------------------------------------------------------------------------------------------------------------------------------------------------------------------------------------------------------------------------------------------------------------------------------------------------------------------------------------------------------------------------------------------------------------------------------------------------------------------------------------------------------------------------------------------------------------------------------------------------------------------------------------------------------------------------------------------------------------------------------------------------------------------------------------------------------------------------------------------------------------------------------------------------------------------------------------------------------------------------------------------------------------------------------------------------------------------------------------------------------------------------------------------------------------------------------------------------------------------------------------------------------------------------------------------------------------------------------------------------------------------------------------------------------------------------------------------------------------------------------------------------------------------------------------------------------------------------------------------------------------------------------------------------------------------------------------------------------------------------------------------------------------------------------------------------------------------------------------------------------------------------------------------------------------------------------------------------------------------------------------------------------------------------------------------|----|--------------|---------------------------|------------------|
| Variable                   | Distribution                                                                                                                                                                                                                                                                                                                                                                                                                                                                                                                                                                                                                                                                                                                                                                                                                                                                                                                                                                                                                                                                                                                                                                                                                                                                                                                                                                                                                                                                                                                                                                                                                                                                                                                                                                                                                                                                                                                                                                                                                                                                                                                  |    | Limit        | Quantity of Interest      | Unit             |
| $U_{\sim}$                 | Weibull                                                                                                                                                                                                                                                                                                                                                                                                                                                                                                                                                                                                                                                                                                                                                                                                                                                                                                                                                                                                                                                                                                                                                                                                                                                                                                                                                                                                                                                                                                                                                                                                                                                                                                                                                                                                                                                                                                                                                                                                                                                                                                                       |    | State | <del></del>               | ~~~              |
| &~                         | Weldun                                                                                                                                                                                                                                                                                                                                                                                                                                                                                                                                                                                                                                                                                                                                                                                                                                                                                                                                                                                                                                                                                                                                                                                                                                                                                                                                                                                                                                                                                                                                                                                                                                                                                                                                                                                                                                                                                                                                                                                                                                                                                                                        |    |              |                           | [ m ]     |
| $Yaw_{mis}$                | Uniform                                                                                                                                                                                                                                                                                                                                                                                                                                                                                                                                                                                                                                                                                                                                                                                                                                                                                                                                                                                                                                                                                                                                                                                                                                                                                                                                                                                                                                                                                                                                                                                                                                                                                                                                                                                                                                                                                                                                                                                                                                                                                                                       |    | SLS          | Surge                     |                  |
| $\overset{\sigma_u}{\sim}$ | Log Normal                                                                                                                                                                                                                                                                                                                                                                                                                                                                                                                                                                                                                                                                                                                                                                                                                                                                                                                                                                                                                                                                                                                                                                                                                                                                                                                                                                                                                                                                                                                                                                                                                                                                                                                                                                                                                                                                                                                                                                                                                                                                                                                    |    |              | Pitch              | [°]              |
| $H_{s_{\infty}}$           | Weibull                                                                                                                                                                                                                                                                                                                                                                                                                                                                                                                                                                                                                                                                                                                                                                                                                                                                                                                                                                                                                                                                                                                                                                                                                                                                                                                                                                                                                                                                                                                                                                                                                                                                                                                                                                                                                                                                                                                                                                                                                                                                                                                       | ~~ |              | Tower Top Acceleration fa | $[m/s^2]_{\sim}$ |
| $W_{dir}$                  | Von Mises                                                                                                                                                                                                                                                                                                                                                                                                                                                                                                                                                                                                                                                                                                                                                                                                                                                                                                                                                                                                                                                                                                                                                                                                                                                                                                                                                                                                                                                                                                                                                                                                                                                                                                                                                                                                                                                                                                                                                                                                                                                                                                                     |    |              | Tower Top Acceleration ss | $[m/s^2]_{\sim}$ |
| ****                       |                                                                                                                                                                                                                                                                                                                                                                                                                                                                                                                                                                                                                                                                                                                                                                                                                                                                                                                                                                                                                                                                                                                                                                                                                                                                                                                                                                                                                                                                                                                                                                                                                                                                                                                                                                                                                                                                                                                                                                                                                                                                                                                               |    | ULS          |                           | [kN]             |
| $T_{\mathcal{R}}$          | Log Normal                                                                                                                                                                                                                                                                                                                                                                                                                                                                                                                                                                                                                                                                                                                                                                                                                                                                                                                                                                                                                                                                                                                                                                                                                                                                                                                                                                                                                                                                                                                                                                                                                                                                                                                                                                                                                                                                                                                                                                                                                                                                                                                    |    |              | Pontoon Bending           |                  |
| ~K                         | L Service Control of the Control of |    | FLS          |                           | [Mpa]            |
| $R_{f_{\sim}}$             | Uniform                                                                                                                                                                                                                                                                                                                                                                                                                                                                                                                                                                                                                                                                                                                                                                                                                                                                                                                                                                                                                                                                                                                                                                                                                                                                                                                                                                                                                                                                                                                                                                                                                                                                                                                                                                                                                                                                                                                                                                                                                                                                                                                       |    |              | Mooring Line 1 DEL        | 7000             |
| ~~~                                                                                                                                                                                                                                                                                                                                                                                                                                                                                                                                                                                                                                                                                                                                                                                                                                                                                                                                                                                                                                                                                                                                                                                                                                                                                                                                                                                                                                                                                                                                                                                                                                                                                                                                                                                                                                                                                                                                                                                                                                                                                                                          |    |              |                           | [Mpa]            |
| $D_{buoy}$                 | Uniform                                                                                                                                                                                                                                                                                                                                                                                                                                                                                                                                                                                                                                                                                                                                                                                                                                                                                                                                                                                                                                                                                                                                                                                                                                                                                                                                                                                                                                                                                                                                                                                                                                                                                                                                                                                                                                                                                                                                                                                                                                                                                                                       |    |              | Mooring Line 2 DEL        |                  |

Time domain simulations are performed with the Hawe2-HAWC2 tool (Horizontal Axis Wind Turbine Simulation Code 2nd Generation (Bischoff Kristiansen, 2022)), using six random seeds per sample point, three for the wind turbulence and two for the waves. This results in a total of 20400 Hawe2-HAWC2 time-domain simulation data points. The resulting 20400 time series are post-processed to obtain the loads, responses, and DELs on the structure, which are summarized in Table 7. The 90 % quantiles of the time series are used to further assess the ULS, to capture the high load/response of the system without being overconservative. The short-term DELs are computed for tower base, mooring lines, and blade roots mooring lines by applying rainflow counting and the Palmgren-Miner's rule (iec, 2019) as in eq. Equation 4 below, where the  $S_i$  is the load/stress amplitude of a number of cycles  $n_i$  in the  $i^{th}$  bin, and m is the Wöhler exponent. The relationship between stress amplitudes and number of cycles to failure is derived from a standard SN curve where  $N = QS^{-m}$ .  $n_{ref}$  is the number of equivalent cycles over a reference period (e.g., setting  $n_{ref}$  to 600 for a 10-minute simulation results in 1 Hz-equivalent loads), and K is the total number of bins for DEL computation.

$$S_{eq} = \left[\frac{\sum_{i=1}^{K} n_i S_i^m}{n_{ref}}\right]^{\frac{1}{m}} \tag{4}$$

$$DEL_{Lifetime} = \left[ \frac{N_{Lifetime}}{n_{ref}M} \sum_{i=1}^{M} S_{eq,i}^{m} \right]^{\frac{1}{m}}$$
 (5)

Lifetime DEL is estimated by direct sampling from the joint environmental probability distribution and obtaining surrogate model responses  $S_{eq.i}$ . The DEL summation is done using Equation 5, following the same notation as in Equation 4. Additional variables are defined as follows:  $N_{Lifetime}$  is the total number of short-term periods over the lifetime of the system,  $n_{ref}$  is the number of cycles for the 1 Hz equivalent simulation length and M is the number of different environmental conditions used to compute  $DEL_{Lifetime}$ . M is defined with a convergence study<del>considering different numbers of environmental conditions, and it is selected as , where we computed  $DEL_{Lifetime}$  for different values of M. Convergence was achieved after 50000 samples.</del>

395

400

405

410

420

425

The DELs are estimated only for the operational range of the turbine. Note that Equation 5 holds when the lifetime DEL are computed by drawing a number of random samples equal to the total number of short-term periods corresponding to the operational life of the system (i.e., the computation effectively simulates the entire lifetime) At this sample size, convergence means we can conclude that the sample is representative enough of the full design space, and we can consider that each environmental condition sample has equal weight. Therefore, there is no need for further scaling of environmental conditions based on their joint probabilities. This approach is convenient as it does not require probability weighting (each Monte Carlo sample is assumed equally likely), but it requires an efficient surrogate model in order to be computationally feasible. The DELs are estimated only for the operational range of the turbine. The DEL computation makes use of an SN curve with an intercept of  $Q = 6.0 \cdot 10^{10}$  and slope of m = 3.0 (Wöhler exponent), selected from the recommended values in DNV (2021) for studless chain mooring in corrosive environment. After estimating  $DEL_{Lifetime}$ , one can define the fatigue limit state for the mooring lines as below in Equation 6:

$$g_{Fatigue}(X) = \Delta - \frac{n_{eq}DEL_{Lifetime}^{m}}{Q}$$
 (6)

where  $\Delta$  is the fatigue damage limit for the material. In this work, the limit state is considered deterministic, and hence  $\Delta$  is taken as 1.

Input features for the surrogate model Variable Distribution U Weibull  $U_{dir}$  von Mises $\sigma_u$  Log Normal  $H_s$  Weibull  $W_{dir}$  von Mises  $T_p$  Log Normal  $R_f$  Uniform  $D_{buoy}$  Uniform

Output features for the surrogate model Limit State Quantity of Interest Unit Surge mPitch Tower Top Acceleration fa  $[N/s^2]$  Tower Top Acceleration ss  $[N/s^2]$  Pontoon Bending kNMooring Line DEL Mpa

A sensitivity analysis is conducted by considering six different cases for different seed configurations and four different simulation lengths to identify the required simulation length and number of seeds for a feasible DOE. Considered cases The cases considered are presented in Table 8. Generally, 10-minute simulations are enough for representing turbulence characteristics, and it is important to separate mean wind conditions from turbulent fluctuations (Burton et al., 2011), but a longer simulation length is vital to capture low-frequency dynamics/loads on the floating structure, which is essential for the floating wind turbines (due to lower surge natural frequencies, especially). For this sensitivity analysis, only a 12 m/s wind speed is considered, and the environmental conditions are selected from the distribution defined in Equation 3. As a measure of control, the median values of each simulation are used, and the percent root mean square (RMS) error is calculated based on six wind and three

wave seeds. Considering the findings from this part, it is decided to use three wind seeds and two wave seeds with a 1400 s simulation length, including a 200 s transient period. The results are presented in Figue 5. This analysis is case dependent, and one should consult the relevant design guidelines such as (DNV, 2014; Veritas, 2010; iec, 2019).

Table 8. Cases considered for the wind seed, wave seed, and simulation length.

| Case   | Wind Seed | Wave Seed | Simulation Length [s] |
|--------|-----------|-----------|-----------------------|
| Case 1 | 3         | 1         | 800, 1400, 2000, 3800 |
| Case 2 | 3         | 2         | 800, 1400, 2000, 3800 |
| Case 3 | 4         | 2         | 800, 1400, 2000, 3800 |
| Case 4 | 4         | 3         | 800, 1400, 2000, 3800 |
| Case 5 | 5         | 3         | 800, 1400, 2000, 3800 |
| Case 6 | 6         | 3         | 800, 1400, 2000, 3800 |

Only a 12 m/s wind speed is considered for the sensitivity analysis, and the environmental conditions are selected from the distribution defined in Equation 3. As a measure of variability, the percent error is calculated based on the median value of six wind and three wave seeds for each time series. Our reference for the error comparison is the case with the highest number of seeds (Six wind and three wave realizations) for each simulation length. The results of this sensitivity analysis are presented for three quantities of interest, including the mooring line tension, tower base moment, and the thrust force.

The results of the sensitivity study are presented in Figure 5. The reader should interpret these results in light of the tradeoff between computational cost and accuracy. Statistically, using more realizations can increase the accuracy of postprocessing; however, we may also lose information on the floater's low-frequency response if the simulation length is not sufficiently long. So, using the results in Figure 5, the readers can select the tradeoff between different numbers of wind and wave realizations for obtaining sufficiently accurate statistics for a selected simulation length. This analysis is case dependent, and one should consult the relevant design guidelines such as (DNV, 2014; Veritas, 2010; iec, 2019). Considering the findings from the sensitivity study, we decided to use three wind seeds and two wave seeds with a 1400-second simulation length, including a 200-second transient period. We built our deterministic surrogate model based on the median of outputs from random seeds for each environmental case.

**3.5.2 Implementing the Surrogate Model**

435

Surrogate model parameters (i.e., hyperparameters) should be tuned for the specific dataset, and there are several available methods for hyperparameter optimization. In this work, we opted to use Bayesian optimization. Bayesian optimization requires fewer iterations and converges to better optimal solutions in less time than traditional hyperparameter tuning algorithms, such as grid search and random search (Snoek et al., 2012). For the surrogate model fitting, we split the input dataset into test and training sets (20 % test and 80 % training), and used the test dataset to validate the model. The FNN architecture consists of two hidden layers with Rectified Linear Unit (RELU) activation functions, and one output layer with a single neuron with a linear activation function as required for regression tasks. Due to efficient gradient propagation, improved computational costs,

Figure 5. Percent Absolute Error absolute error for Different Number different number of Seeds wave and wind seeds with respect to simulation length

and good convergence properties (Alba et al., 2021), the RELU activation is preferred. The model accuracy is validated using different error metrics, including RMSE and the coefficient of determination (R-squared value).

**3.6 Second Step: Optimization Based on Global Structural Response Computed in Time Domain**

Global system performance is preferred when investigating different floater characteristics and evaluating the limit states. The global limit states considered in this study are divided into three main types: ultimate limit state (ULS), fatigue limit state (FLS), and serviceability limit state (SLS). SLS is defined We defined SLS as the maximum allowable floater offset that avoids to avoid dynamic cable damage; FOWT pitch angle, which determines system stability and power generation; and, FWT pitch angle to determine system stability, and limit power loss. Finally, the nacelle acceleration limit, which prevents damage to the drivetrain and preserves its fatigue life set for a safe operation of the turbine. ULS is defined as a maximum mooring line

tensile load, floater pontoon buckling, and tower base bucklingand floater pontoon bending. FLS is defined in terms of the lifetime fatigue of the structure's mooring lines. The design optimization in the time domain is conducted for multiple limit states simultaneously, while separate surrogate models are trained for each limit state based on the same training dataset. In Table 9, the global limit states for the second stage of the optimization are presented. SLS are given as  $g_{11}(x_1)$  for dynamic surge motion,  $g_{12}(X)$  for the dynamic pitch angle,  $g_{13}(X)$  is for the maximum nacelle acceleration.  $g_{14}(X)$  static pontoon bending  $g_{15}(X)$  and maximum mooring line tension  $g_{16}(X)$  are ULS and mooring line fatigue lifetime fatigue for mooring line 1 and mooring line 2 defined as  $g_{16}(X)$  and  $g_{17}(X)$ .

Table 9. Design Constraints for the first part of the optimization where  $\delta_1$  is the surge motion,  $\delta_2$  is the dynamic pitch angle,  $a_{nacelle}$  is the tower top acceleration, and  $T_{max}$  is the maximum mooring line tension predicted by the surrogate model.  $T_{cr}$  is the mooring line tension capacity and  $\sigma_{design}$  is the bending load on the pontoons.  $D_{Life,moor1}$  and  $D_{Life,moor2}$  are the lifetime fatigue capacities of the mooring lines.

| Constraint              | Description                         |                                                              | Unit                            |  |  |
|-------------------------|-------------------------------------|--------------------------------------------------------------|---------------------------------|--|--|
|                         | Serviceability Limit States         |                                                              |                                 |  |  |
| $g_{11}(X)$             | Dynamic Surge offset                | $\delta_1 - 40$                                              | m                               |  |  |
| $g_{12}(X)$             | Dynamic Pitch angle                 | $\delta_2 - 10$ (Dou et al., 2020; Leimeister et al., 2020b) | 0                               |  |  |
| $g_{13}(X)$             | Nacelle acceleration side side (ss) | $a_{nacelle} - 1.962$ (Leimeister et al., 2020b)             | m/s 2                |  |  |
| $g_{14}(X)$             | Nacelle acceleration fore aft (fa)  | $a_{nacelle} - 1.962$ (Leimeister et al., 2020b)             | $\underset{\sim}{\text{m/s}^2}$ |  |  |
|                         | Ultimate Limit States               |                                                              |                                 |  |  |
| $g_{14}(X)$ $g_{15}(X)$ | Mooring line 1 tension              | $T_{max} - T_{cr}$                                           | kN                              |  |  |
| $g_{15}(X)$ $g_{16}(X)$ | Static pontoon bending              | $\sigma_{design} - \sigma_{cr}$                              | <del>MPa</del> kN  €            |  |  |
| Fatigue Limit States    |                                     |                                                              |                                 |  |  |
| $g_{16}(X)$ $g_{17}(X)$ | Lifetime fatigue for mooring line 1 | $D_{Life,moor1} - 1$                                         | -                               |  |  |
| $g_{17}(X) g_{18}(X)$   | Lifetime fatigue for mooring line 2 | $D_{Life,moor2} - 1$                                         | -                               |  |  |

The dynamic optimization constraints are selected based on the system dynamics and production limits, considering the practices seen in literature. Dynamic surge motion is restricted, considering the maximum dynamic floater offset determined by the designer and the mooring line design characteristics. In literature, there are various ways of selecting this value. Pollini et al. (2023) used 25 % of the water depth as dynamic surge displacement criteria, where (Leimeister et al., 2020b) used 20 % of the water depth as the static surge displacement criteria and then minimized the dynamic surge motion within the optimization. A strict nacelle acceleration constraint is implemented for operational reasons to avoid lubrication problems for the sensitive nacelle components (Leimeister et al., 2020b). Constraints for the ULS are defined based on the tension failure of the mooring lines and tower base yielding, which are critical for the integrity of the FWT assets. The final limit states (FLS) considered are those related to the fatigue lifetime of the mooring lines, which is critical for the stability and station-keeping of the entire system, as well as preventing fatigue failure during the FWT lifetime.

**4 Results**

495

500

505

510

**4.1 Low Cost Design Evaluation: First Step**

This section presents the results of the analytical design constraint evaluation. The boundaries for each design constraint considered can be seen. The initial step of our approach is to reduce the design space by evaluating the analytical design constraints. As discussed in Section 3.1, these include natural frequencies, stability, and static pitch angle under rated wind speed. The results are presented in Figure 6-, where subfigures 6a to 6f present the order of constraint evaluation. In Figure 6, the samples shown are infeasible designs, and different colors indicate the active constraints that render the designs infeasible.

In Figure 6a, surge natural frequency constraint (g1(X)) eliminates the designs at the lower left corner of the DOE. Pitch natural frequency lower (g2(X)) and upper (g3(X)) constraints narrows the feasible design space further. Evaluating tower natural frequency (g4(X)) and system stability (g5(X)) constraints causes fewer infeasible design points compared to pitch natural frequency and static pitch angle (g6(X)) constraints. When all analytical design constraints are considered as in Figure 6f, the final feasible design space is bounded by the static pitch angle and pitch natural frequency constraint. This result is concept and design variable-dependent, and different shapes can be obtained with different variables.

Surge natural frequency constraint Pitch natural frequency constraint (low)

Pitch natural frequency constraint (high) Tower natural frequency constraint

Stability Constraint Static Pitch Angle

Analytical design constraints evaluated sequentially for the initial DOE screening After running the analytical design constraint evaluation, the Pareto fronts of the feasible design space are defined to identify its boundaries. The exact boundary fit is offset to include potentially feasible design points that may not be captured by the relatively crude sampling used in the first optimization stage. We observe that in the present case the Pareto fronts can be approximated by a power law. The boundaries of a design space parameterized with a power law decay function can be seen in Figure 7a, and the resulting DOE for the time domain evaluation can be seen in Figure 7b —with LCOE values. The LCOE values have a decreasing trend towards longer floater radius and smaller outer column diameter.

**4.2 Time Domain Design Evaluation: Second Step**

The second phase of the optimization includes results from the global limit states. Figure 8 presents the reference design and optimized design geometries side-by-side, and Figure 9 presents the iteration history of the design variables and objective function. Compared to the baseline design, floater radius  $x_1$  has an increasing trend with the number of evaluations, and the outer column diameter  $x_2$  decreases. Considering our objective function LCOE, this results in a lower objective function value. As shown in Figure 10, the design driving constraints are mooring line fatigue active constraints are the mooring line 1 fatigue  $(g_{17}(X))$ , which is in the direction of loading, and the lower boundary from the first step of the design optimization. Design constraints, such as mooring 2 fatigue  $(g_{18}(X))$ , mooring 1 tension  $(g_{15}(X))$ , and tower top accelerations, do not affect our optimal solution. During the initial optimization iterations, the dynamic pitch angle constraint  $(g_{12}(X))$  was active, and it became inactive with increasing floater radius and decreasing outer column diameter. As presented in Figure 6, the lower

Figure 6. Analytical design constraints evaluated sequentially for the initial DOE screening

(a) Boundary fit to the feasible design space (b) Latin hypercube grid of the feasible design space

Figure 7. Refined design space after the analytical limit state evaluation

boundary defining our feasible design space is determined by the mean pitch angle at rated wind speed. Mooring line design has effects on both active constraints. By changing the mooring line parameters, one may further improve the design, where co-design approaches might be implemented.

(b) Optimized Design

Figure 8. Comparison of reference and optimized design

The resulting design has 60.18 m floater radius and 10.3 m buoyancy outer column diameter with LCOE of 176.9 €/MWh, which is a 3.7 % decrease from the nominal value. Key load responses of the optimized design are also compared to the baseline

design as presented in Figure 12. The simulations for this comparison are performed using HAWC2 (Bischoff Kristiansen, 2022), and the same wind and wave seeds are used to eliminate seed uncertainty. Median time series statistics are used for the 2000s simulations. The simulations are performed for the operational range with NTM turbulence. Figure 12 presents that, regarding aerodynamics, both designs have the same responses for the generated power and blade root moment. There are no visible differences for the hydrodynamic loads/responses. The difference is higher for the above-rated region for the mooring line tension. As it is presented in Figure 11, pitch response is higher compared to the reference design. The response is still within the allowable limit of 10 degrees. Considering those findings, we can say that both designs show similar behavior, but for further design verification, additional simulations and potentially higher-fidelity analysis should be performed. Reference and optimized designs are also compared considering the AEPs for the South Brittany Site. The AEP of the reference design is calculated as 85.93 GWh and the AEP of the optimized design is estimated at 85.74 GWh which is 0.2211 % lower than the reference design. The difference is mainly due to the larger pitch response of the optimized floater where this tradeoff should be considered as well.

Figure 9. Design variables and cost values during the iteration for the optimization with mooring line FLS

**5 Discussion**

525

The method presented in this paper has several advantages, including the effective combination of analytical constraints and surrogate modeling to reduce computational complexity. It—With our approach, we obtained a significant decrease in the computational cost of the optimization problem. This is achieved by two important steps: replacing time-domain load simulations with a surrogate model and reducing DOE for surrogate training. Using a surrogate model within the optimization

Figure 10. Reference DesignConstraint values for the iteration with mooring line FLS

Optimized Design Comparison of reference and optimized design

535

process results in a reduction of 98.72 % of simulations to run. Without a surrogate model, about 1.6 million ten-minute simulations should be performed for the optimization process. With our approach, only 20400 aeroelastic simulations should be performed for surrogate model training. A second improvement is due to reducing the DOE area for surrogate model training, whereby, focusing only on the feasible design space, the sampling space for surrogate training is decreased by 89.4 %. Therefore, our approach enables the evaluation of limit states based on high-fidelity time-domain analyses without excessive computational costs. Due to the flexibility of the framework, additional design variables or limit states can be added, which offers a practical balance between conceptual simplicity and detailed accuracy, ideal for iterative optimization processes.

The presented algorithm also faces limitations and challenges. A significant part of the computational cost associated with the use of the framework is due to the data generation required for surrogate model training. HPC simulations are currently the only feasible approach for generating the surrogate training set. In the present study, the simulations were performed on the Sophia HPC cluster owned by the Technical University of Denmark. Depending on the complexity of the aeroleastic model, this phase can be improved further. The computational cost for surrogate model training and for optimization (both the first step

(a) Iteration history—Comparison of power generation for the optimized and baseline design variables

550

555

560

565

(b) Iteration history Comparison of the objective function pitch response for optimized and baseline design

Figure 11. Design variables Comparison of power generation properties of optimized and eost values during the iteration for the optimization with mooring line FLS baseline design.

and the second step as described earlier) is in the order of minutes on a single CPU, which is negligible compared to the training data generation cost. For increasing accuracy of the surrogate model, different sampling techniques can be implemented, the DOE of the surrogate can be enriched incrementally using active learning approach, and additional environmental conditions can be considered.

The cost of generating surrogate model training data has to be weighted weighted against considerations on the accuracy of the resulting surrogate, as well as the representativeness of the design problem. Considering a limited number of design variables can cause neglect of additional influential design variables, such as structural thickness and mooring design properties.

The steel plate thickness is considered constant throughout the floater, which is likely not representative of the final floater design. This should be further investigated in the detailed design phase using FEM (Finite Element Modeling).

The bending limit state is considered only for the static loading case. To indirectly include dynamic effects, an adjustment factor can be implemented within the optimization problem. Within our approach, we developed an algorithm that is suitable for uncertainty propagation within the design optimization process. Currently, we have only considered the environmental uncertainties by sampling from the environmental distributions of the selected site. In an addition to our approach, one can consider the effects of other uncertainties, such as material, load, and model, within the optimization.

Finally, the results obtained from this work demonstrated the potential of surrogate-based optimization methods for meaningful LCOE reduction in FWT designs. They also highlighted the feasibility and accuracy in bridging the gap between conceptual and detailed designs, which indicates significant potential for broader application across different FWT concepts and larger-scale optimization problems. Future work should aim to expand the optimization scope to include more detailed design variables, incorporate robust or reliability-based design optimization to address uncertainties, and evaluate a wider range of optimization algorithms. Additionally, exploring probabilistic surrogate models could enhance the ability to quantify and manage uncertainties inherent in offshore wind conditions and cost estimations.

Figure 12. Constraint values Median key load response comparison for the iteration optimized design with mooring line FLSthe reference case using HAWC2 simulations.

**6 Conclusion**

[revised manuscript text omitted]